# Proper chromosome alignment depends on BRCA2 phosphorylation by PLK1

Åsa Ehlén [1,2], Charlotte Martin [1,2,8], Simona Miron[3,8], Manon Julien[3,4,8], François-Xavier Theillet [3], Virginie Ropars[3], Gaetana Sessa[1,2], Romane Beaurepere[1,2], Virginie Boucherit[1,2], Patricia Duchambon[5,6], Ahmed El Marjou[5,7], Sophie Zinn-Justin[3,9✉] & Aura Carreira[1,2,9✉]

The BRCA2 tumor suppressor protein is involved in the maintenance of genome integrity through its role in homologous recombination. In mitosis, BRCA2 is phosphorylated by Polo-like kinase 1 (PLK1). Here we describe how this phosphorylation contributes to the control of mitosis. We identify a conserved phosphorylation site at T207 of BRCA2 that constitutes a bona fide docking site for PLK1 and is phosphorylated in mitotic cells. We show that BRCA2 bound to PLK1 forms a complex with the phosphatase PP2A and phosphorylated-BUBR1. Reducing BRCA2 binding to PLK1, as observed in *BRCA2* breast cancer variants S206C and T207A, alters the tetrameric complex resulting in unstable kinetochore-microtubule inter-actions, misaligned chromosomes, faulty chromosome segregation and aneuploidy. We thus reveal a role of BRCA2 in the alignment of chromosomes, distinct from its DNA repair function, with important consequences on chromosome stability. These findings may explain in part the aneuploidy observed in *BRCA2*-mutated tumors.

[1] Institut Curie, PSL Research University, CNRS, UMR3348, F-91405 Orsay, France. [2] Paris Sud University, Paris-Saclay University CNRS, UMR3348, F-91405 Orsay, France. [3] Institute for Integrative Biology of the Cell (I2BC), CEA, CNRS, Univ Paris-Sud, Université Paris-Saclay, Gif-sur-Yvette, Cedex, France. [4] Department of Biology, École Normale Supérieure, 94230 Cachan, France. [5] Protein Expression and Purification Core Facility, Institut Curie, 26 rue d'Ulm, 75248 Paris, Cedex 05, France. [6] INSERM U1196, 91405 Orsay, Cedex, France. [7] CNRS UMR144, 12 rue Lhomond, 75005 Paris, France. [8] These authors contributed equally: Charlotte Martin, Simona Miron, Manon Julien. [9] These authors jointly supervised: Sophie Zinn-Justin, Aura Carreira. ✉email: sophie.zinn@cea.fr; aura.carreira@curie.fr

The BRCA2 tumor suppressor protein plays an important role in DNA repair by homologous recombination (HR)[1,2] that takes place preferentially during S/G2 phases of the cell cycle[3]. BRCA2 has also emerging functions in mitosis, for example, at the kinetochore, it forms a complex with BUBR1[4,5], a protein required for kinetochore–microtubule attachment and a component of the spindle assembly checkpoint (SAC)[6,7]. These two activities of BUBR1 involve different partners and are functionally distinct[8,9]. BRCA2 has been proposed to contribute to BUBR1 SAC activity[4,10], although due to confounding results in the BUBR1 interaction site in BRCA2, it is unclear if this interaction is direct[4,5]. At the end of mitosis, BRCA2 localizes to the midbody and assists cell division by serving as a scaffold protein for the central spindle components[11–13]. In mitosis, BRCA2 is a target of phosphorylation by PLK1 both in its N-terminal region[14,15] and in its central region[15], although the functional role of these phosphorylation events remains unclear.

PLK1 is a master regulator of the cell cycle that is upregulated in mitosis[16,17]. Among other functions, PLK1 directly binds and phosphorylates BUBR1 at several residues including the two tension-sensitive sites S676[18] and T680[19] in prometaphase allowing the formation of stable kinetochore–microtubule attachments. This activity needs to be tightly regulated to ensure proper alignment of the chromosomes at the metaphase plate[8,9,18]. The kinase activity of Aurora B is essential to destabilize erroneous kinetochore–microtubule interactions[20] whereas the phosphatase PP2A protects initial kinetochore–microtubule interactions from excessive destabilization by Aurora B[21]. This function is achieved through the interaction of PP2A-B56 subunit with BUBR1 phosphorylated at the Kinetochore Attachment and Regulatory Domain (KARD) motif (including residues S670, S676, and T680)[19]. Thus, the interplay between PLK1, BUBR1, Aurora B, and PP2A is necessary for the formation of stable kinetochore–microtubule attachments.

PLK1 is recruited to specific targets via its Polo-box domain (PBD)[22]. PBD interacts with phosphosites characterized by the consensus motif S-[pS/pT]-P/X[23]. These phosphosites are provided by a priming phosphorylation event, usually mediated by CDK1 or other proline-directed kinases[17]; however, there is also evidence that PLK1 itself might create docking sites ("self-priming") during cytokinesis[24,25].

Several BRCA2 sites have been suggested as phosphorylated by PLK1 in mitosis, some of which belong to a cluster of serines and threonines located in BRCA2 N-terminus around residue S193[14]. We set out to investigate which of these sites are phosphorylated by PLK1, and to reveal whether these phosphorylation events play a role in the regulation of mitotic progression. Here, we demonstrate that the two conserved residues S193 and T207 are phosphorylated by PLK1, and that phosphorylated BRCA2-T207 is a bona fide docking site for PLK1. By investigating the phenotype of BRCA2 missense variants that limit the phosphorylation of BRCA2-T207, we reveal an unexpected role for BRCA2 in the alignment of chromosomes at the metaphase plate. We demonstrate that phosphorylation of BRCA2-T207 by PLK1 facilitates the formation of a complex between BRCA2-PLK1-pBUBR1 and the phosphatase PP2A. A defect in this function of BRCA2 is manifested in chromosome misalignment, chromosome segregation errors, mitotic delay and aneuploidy, leading to chromosomal instability.

## Results

### Breast cancer variants alter PLK1 phosphorylation of BRCA2.
Several missense variants of uncertain significance (VUS) identified in *BRCA2* in breast cancer patients are located in the N-terminal region predicted to be phosphorylated by PLK1

(around S193) (Breast information core (BIC)[26] and BRCA-Share[27]), summarized in Supplementary Table 1. To find out if any of these variants affected PLK1 phosphorylation in this region, we purified fragments comprising amino acids 1 to 250 of BRCA2 (hereafter BRCA2$_{1–250}$) from human embryonic kidney cells (HEK293T) and used an in vitro kinase assay to assess the phosphorylation by PLK1 of the fragments containing either the WT sequence, the different BRCA2 variants M192T, S196N, S206C, and T207A, or the mutant S193A, previously reported to reduce the phosphorylation of BRCA2 by PLK1[14]. As expected, S193A reduced the phosphorylation of BRCA2$_{1–250}$ by PLK1 (Fig. 1a, b). Interestingly, variants T207A and S206C also led to a 2-fold decrease in PLK1 phosphorylation of BRCA2$_{1–250}$ (Fig. 1a, b). In contrast, M192T and S196N did not significantly modify the phosphorylation of BRCA2$_{1–250}$ by PLK1 (Fig. 1a, b). The phosphorylation observed in the BRCA2 fragments is specific of the recombinant PLK1 kinase as it is PLK1 concentration dependent (Supplementary Fig. 1a, b) and when replacing the PLK1-WT by a kinase-dead (PLK1-KD) version of the protein (K82R)[28], purified using the same protocol, or adding a PLK1 inhibitor (BI2536) to the reaction, the phosphorylation of BRCA2$_{1–250}$ decreased significantly (Fig. 1c, lanes 4 and 5 compared to lane 3; Fig. 1d).

Together, these results show that VUS T207A and S206C identified in breast cancer patients impair phosphorylation of BRCA2$_{1–250}$ by PLK1 in vitro.

### BRCA2-T207 is a target of phosphorylation by PLK1.
The reduction of BRCA2 phosphorylation in BRCA2$_{1–250}$ containing T207A and S206C variants suggested that these residues could be targets for PLK1 phosphorylation. We investigated this possibility by following the PLK1 phosphorylation kinetics of two overlapping fragments of BRCA2 N-terminus comprising S206 and T207 (hereafter BRCA2$_{48–218}$ and BRCA2$_{190–284}$) using Nuclear Magnetic Resonance (NMR) spectroscopy (Fig. 2a). Together, these fragments cover a large N-terminal region of BRCA2 including the cluster of conserved residues around S193 (from amino acid 180 to amino acid 210; Supplementary Fig. 1c). NMR analysis allows residue-specific quantification of $^{15}N$-labeled peptide phosphorylation. Figure 2b shows superimposed $^{1}H$-$^{15}N$ HSQC spectra of BRCA2$_{48–218}$ and BRCA2$_{190–284}$ before (black) and after (red) phosphorylation with recombinant PLK1. Analysis of these experiments revealed phosphorylation of S193 and of three other phosphosites, including T207, by PLK1 in the BRCA2 region from amino acid 48 to amino acid 284. Interestingly, while T219 and T226 conservation is poor, T207 and S193 are conserved from mammals to fishes (Supplementary Fig. 1c) suggesting that both residues are important for BRCA2 function.

### BRCA2 variant T207A alters PLK1 phosphorylation kinetics.
Having identified T207 as a target of phosphorylation of PLK1, we next compared the residue-specific phosphorylation kinetics in the polypeptide WT BRCA2$_{48–218}$ containing the variant T207A that displayed reduced overall phosphorylation (Fig. 1a, b). (The production of a $^{15}N$-labeled recombinant fragment comprising S206C yielded an insoluble protein precluding NMR analysis). Time-resolved NMR experiments revealed that PLK1 phosphorylates significantly less BRCA2$_{48–218}$ containing the variant T207A than the WT peptide (Fig. 2c). The initial phosphorylation rate of S193 was decreased by a factor of 2 (Fig. 2d), and T207 was, as expected, not phosphorylated, being mutated into an alanine. Similar results were obtained using BRCA2$_{190–284}$ (Supplementary Fig. 2). This NMR analysis is consistent with the in vitro kinase assay performed using the BRCA2$_{1–250}$ fragment

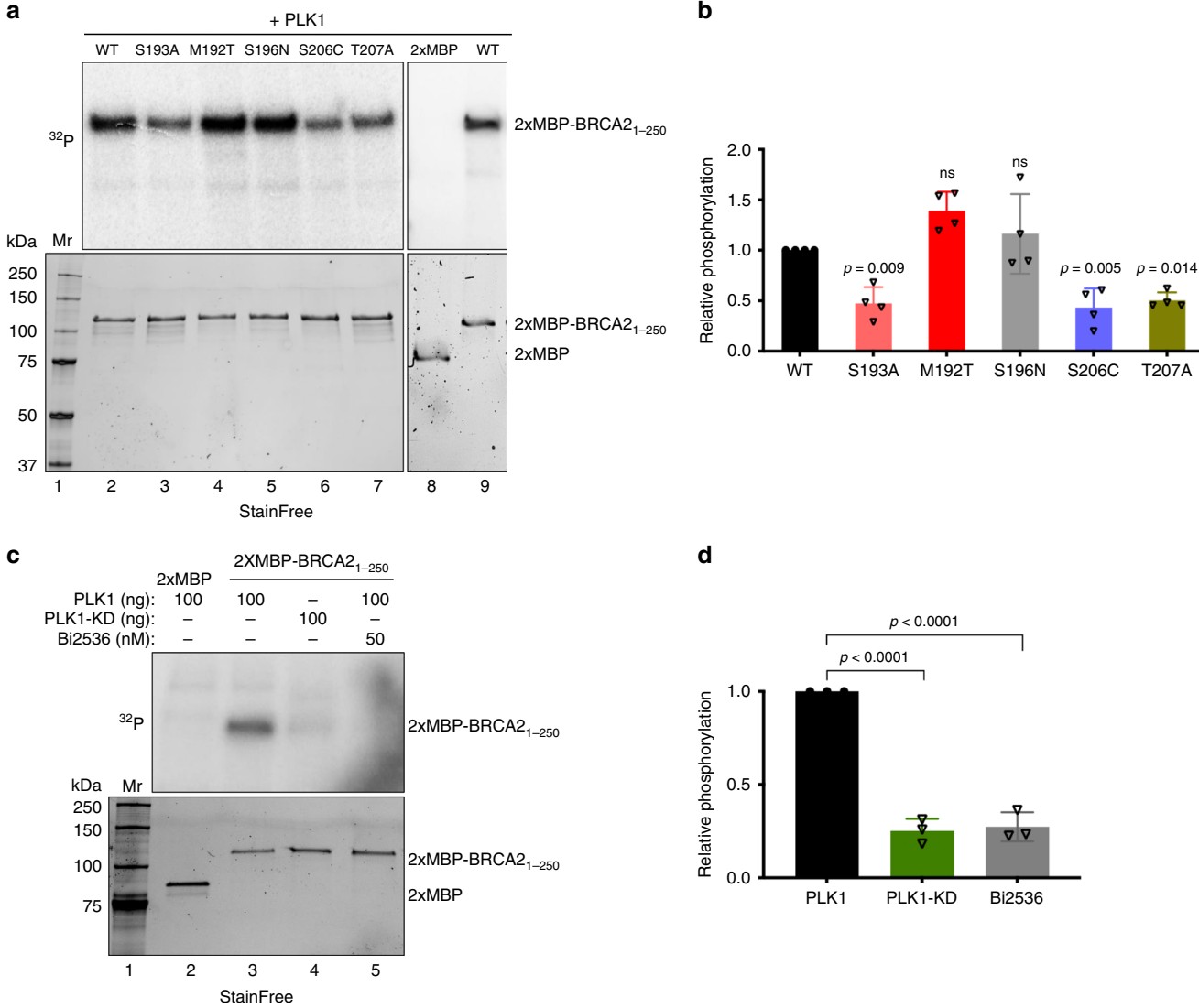

**Fig. 1 BRCA2 VUS alter PLK1 phosphorylation of BRCA2$_{1-250}$. a** PLK1 in vitro kinase assay with BRCA2$_{1-250}$. Top: The polypeptides encompassing 2×-MBP-BRCA2$_{1-250}$ WT or S193A, M192T, S196N, S206C, T207A mutations or the 2XMBP tag were incubated with recombinant PLK1 in the presence of γ$^{32}$P-ATP. The samples were resolved on 7.5% SDS-PAGE and the $^{32}$P-labeled products were detected by autoradiography. Bottom: 7.5% SDS-PAGE showing the input of purified 2xMBP-BRCA2$_{1-250}$ WT and mutated proteins (0.5 μg) used in the reaction as indicated. Mr; molecular weight markers. **b** Quantification of the relative phosphorylation in (**a**). Data in (**b**) are represented as mean ± SD from four independent experiments. **c** PLK1 in vitro kinase assay performed as in (**a**) with recombinant PLK1 or the PLK1 kinase dead K82R mutant (PLK1-KD) together with BRCA2$_{1-250}$ WT as substrate, in the presence or absence of the PLK1 inhibitor BI2536 (50 nM) in the kinase reaction buffer. Mr; molecular weight markers. **d** Quantification of the relative phosphorylation in (**c**). Data in (**d**) are represented as mean ± SD from three independent experiments. **b, d** One-way ANOVA test with Dunnett's multiple comparisons test was used to calculate statistical significance of differences (the p-values show differences compared to WT (**b**) or PLK1 (**d**); ns (non-significant)). Source data are available as a Source Data file.

purified from human cells (Fig. 1), in which T207A reduces the phosphorylation of BRCA2$_{1-250}$ fragment by PLK1.

**Variants T207A and S206C reduce the interaction of BRCA2 with PLK1.** The finding that T207 is efficiently phosphorylated by PLK1 in BRCA2$_{48-218}$ and BRCA2$_{190-284}$ (Fig. 2b) together with the observation that T207A mutation causes a global decrease in the phosphorylation of these fragments (Fig. 2c; Supplementary Fig. 2) and the prediction that T207 is a docking site for PLK1$_{PBD}$ binding[23] made us hypothesize that T207 might be a "self-priming" phosphorylation event required for the interaction of PLK1 with BRCA2 at this site. If so, the variants that reduce phosphorylation of T207 by PLK1 would be predicted

to alter PLK1$_{PBD}$ binding. To test this hypothesis, we examined the interaction of PLK1 with the VUS-containing polypeptides. We overexpressed 2xMBP-BRCA2$_{1-250}$ constructs carrying these variants in U2OS cells to detect the endogenous PLK1 that co-immunoprecipitates with 2xMBP-BRCA2$_{1-250}$ using amylose pull-down. As expected, overexpressed BRCA2$_{1-250}$ was able to interact with endogenous PLK1 from mitotic cells but not from asynchronous cells (predominantly in G1/S) where the levels of PLK1 are reduced (Fig. 3a, lane 2 compared to lane 1). Furthermore, the variants T207A and S206C showed a weaker interaction with PLK1 than the WT protein (Fig. 3a, pull-down lanes 4, 6 compared to lane 2, Fig. 3b) despite the protein levels of PLK1 remaining unchanged (Fig. 3a, compare PLK1 input lanes 4 and 6 to lane 2). In contrast, the effect of M192T and S196N on the

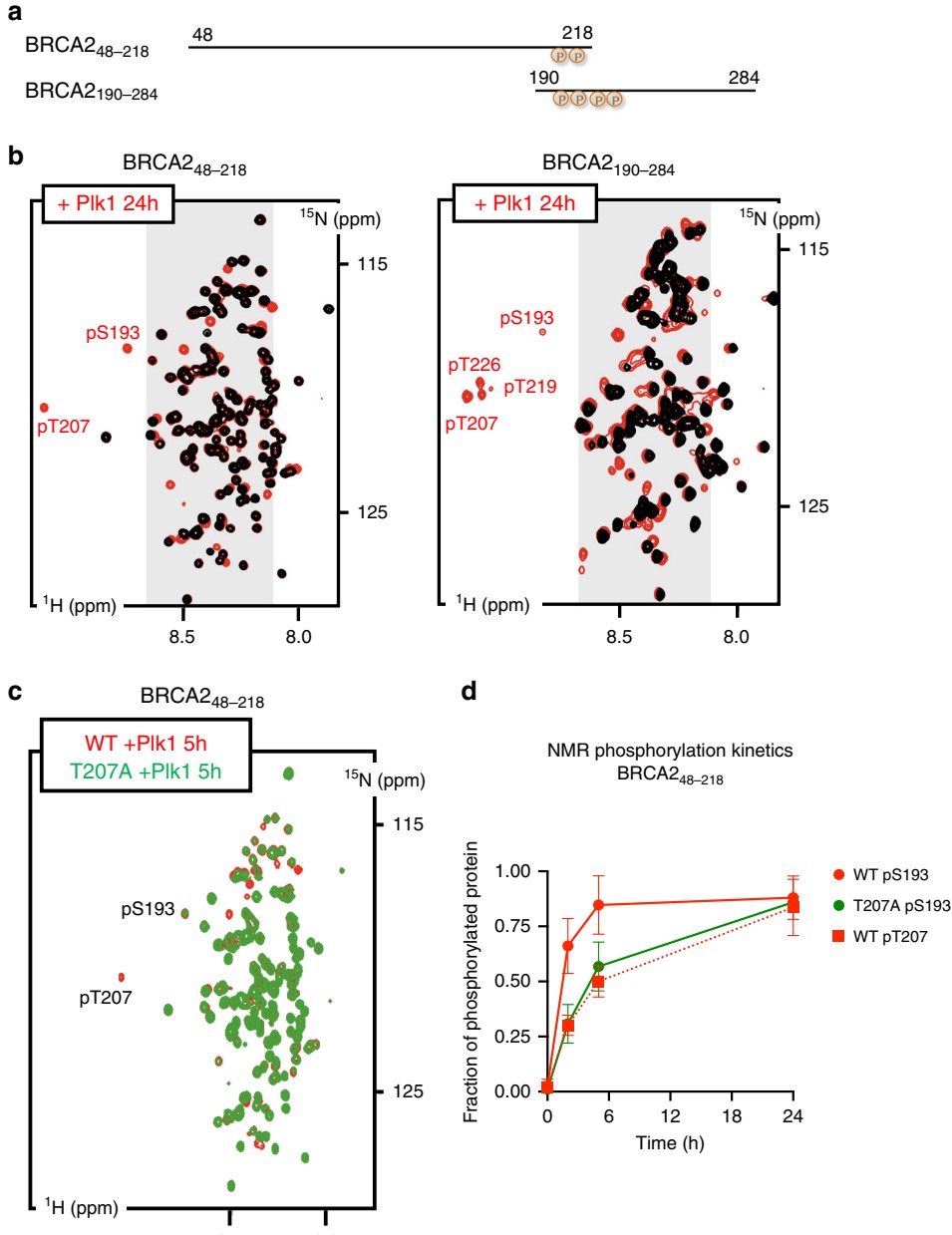

**Fig. 2 PLK1 phosphorylates T207 in BRCA2₄₈₋₂₁₈ and BRCA2₁₉₀₋₂₈₄.** Phosphorylation of BRCA2$_{48-218}$ and BRCA2$_{190-284}$ by PLK1 as observed by NMR. **a** Schematic view of the two overlapping BRCA2 fragments analyzed by NMR. Residues identified as phosphorylated in (**b**) are indicated. **b** Superposition of the $^1$H-$^{15}$N HSQC spectra recorded before (black) and after (24 h; red) incubation with PLK1. Each spectrum contains one peak per backbone NH group. Most peaks are located in the gray region of the spectrum, indicating that they correspond to disordered residues. Assignment of each peak to a BRCA2 residue was performed using a classical heteronuclear NMR strategy. Peaks corresponding to phosphorylated residues are indicated. **c** Comparison of the phosphorylation kinetics recorded for BRCA2$_{48-218}$ WT and T207A. $^1$H-$^{15}$N HSQC spectra of BRCA2$_{48-218}$WT (red) and T207A (green) recorded 5 h after addition of PLK1 are superimposed to highlight the overall decrease of phosphorylation observed for the mutant compared to the WT. **d** Fraction of phosphorylated protein deduced from the intensities of the peaks corresponding to the non-phosphorylated and phosphorylated residues plotted as a function of time. WT S193 and T207 timepoints are represented by red circles and squares, respectively, while T207A S193 timepoints are represented by green circles. The graph represents data as mean ± SD from three independent experiments. Source data are available as a Source Data file.

interaction was mild (Fig. 3a, compare pull-down lanes 10 and 12 to lane 8, Fig. 3b). These results suggest a self-priming phosphorylation by PLK1 on T207.

To provide further evidence that the PLK1-mediated phosphorylation of BRCA2 favors BRCA2 binding, we performed an in vitro kinase assay with recombinant proteins followed by an amylose pull-down and eluted the bound proteins with maltose. PLK1 was found in the maltose elution with BRCA2$_{1-250}$-WT

demonstrating that PLK1-phosphorylated BRCA2$_{1-250}$ binds to PLK1 (Fig. 3c lane 4, Fig. 3d). In contrast, the fraction of PLK1 in the eluate of BRCA2$_{1-250}$-T207A was substantially reduced (Fig. 3c, lane 8 compared to lane 4, Fig. 3d) indicating that the phosphorylation of T207 is required for efficient binding to PLK1 and confirming our results with cell lysates (Fig. 3a, b). Strikingly, we observed no difference in PLK1 binding between BRCA2$_{1-250}$-WT phosphorylated by PLK1 or its kinase dead mutant (PLK1-

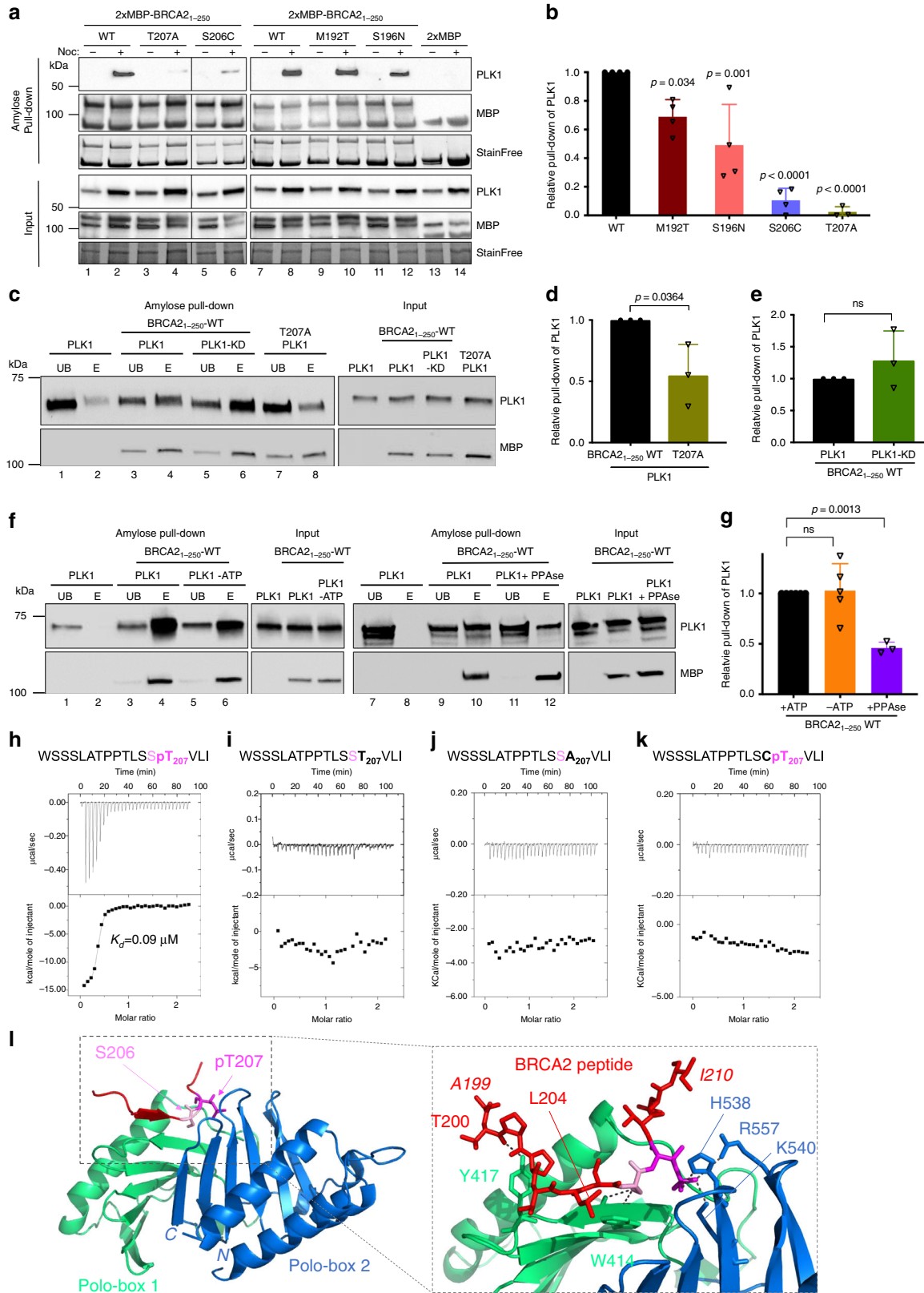

KD; Fig. 3c, lane 6 compared to lane 4, Fig. 3e) or in the absence of ATP (Fig. 3f, lane 6 compared to lane 4, Fig. 3g). However, pre-incubating BRCA2$_{1-250}$-WT with phosphatase before the addition of PLK1 resulted in a 2-fold decrease in the binding to PLK1 indicating that the phosphorylation of BRCA2 is required for the interaction with PLK1 (Fig. 3f lane 12 compared to 10, Fig. 3g).

**T207 is a bona fide docking site for PLK1.** To demonstrate the recognition of pT207 by PLK1, we measured the affinity of recombinant PLK1$_{PBD}$ (the target recognition domain of PLK1) for a synthetic 17 aa peptide comprising phosphorylated T207. Using isothermal titration calorimetry (ITC), we found that recombinant PLK1$_{PBD}$ bound to the pT207 peptide with an

**Fig. 3 BRCA2 variants showing reduced phosphorylation by PLK1 impair PLK1 binding. a** Amylose pull-down of U2OS transiently expressing 2xMBP-BRCA2$_{1-250}$ (WT), the variants (M192T, S196N, S206C, and T207A) or the 2XMBP-tag treated with nocodazole as indicated. 4–15% SDS-PAGE followed by WB using anti-PLK1 and anti-MBP antibodies. StainFree images are used as loading control (cropped image is shown). **b** Quantification of co-immunoprecipitated PLK1 with WT in (**a**), relative to the input levels of PLK1. Results are presented as the fold change compared to WT. The data represent the mean ± SD of three to four independent experiments (WT ($n = 4$), M192T ($n = 4$), S196N ($n = 4$), S206C ($n = 4$), T207A ($n = 3$)). One-way ANOVA test with Dunnett's multiple comparisons test was used to calculate statistical significance of differences ($p$-values compared to WT). **c** PLK1 (or PLK1-KD) as indicated in vitro kinase assay followed by amylose pull-down of BRCA2$_{1-250}$-WT or T207A. 10% SDS-PAGE followed by WB using anti-PLK1 and anti-MBP antibodies. **d, e** Quantification of the PLK1 pull-down in (**c**) relative to the PLK1 levels in the input. Results are presented as the fold change compared to BRCA2$_{1-250}$-WT in (**d**) and PLK1-WT in (**e**). The data represent the mean ± SD of three independent experiments, two-tailed unpaired $t$-test was used to calculate significance of differences (ns (non-significant)). **f** PLK1 in vitro kinase assay followed by amylose pull-down assay as in (**c**) Left panel: with or without ATP as indicated. Right panel: The 2×-MBP-BRCA2$_{1-250}$-WT polypeptide was pre-treated with phosphatase (FastAP) for 1 h before the amylose pull-down (**c**). In (c) and (**f**): UB: unbound, E: eluted. **g** Quantification of the PLK1 pull-down in (**f**) relative to the PLK1 levels in the input. Results are presented as the fold change compared to kinase assay performed with non-phosphatase treated BRCA2$_{1-250}$-WT in the presence of ATP. The data represent the mean ± SD of three to five independent experiments (+ATP ($n = 5$), −ATP ($n = 5$), +PPAse ($n = 3$)), one-way ANOVA test with Dunnett's multiple comparisons test was used to calculate statistical significance of differences ($p$-values compared to +ATP; ns (non-significant)). **h–k** Isothermal Titration Calorimetry (ITC) thermograms showing binding of PLK1$_{PBD}$ to a 17 aa BRCA2 peptide containing (**h**) pT207, (**i**) T207, (**j**) A207, (**k**) C206pT207. Residues S206 and pT207 are highlighted in pink (S206) and magenta (pT207) in the peptide sequences. **l** Left panel: 3D cartoon representation of the crystal structure of PLK1$_{PBD}$ (Polo-box 1 in green and Polo-box 2 in blue) bound to the BRCA2 peptide containing pT207 (in red except for S206 (pink, sticks) and pT207 (magenta, sticks)). Right panel: zoom in on the interface between PLK1$_{PBD}$ and the BRCA2 peptide (from A199 to I210). The amino acids of PLK1$_{PBD}$ and BRCA2 involved in the interaction are highlighted in sticks representation with hydrogen bonds depicted as dark gray dots. Source data are available as a Source Data file.

affinity of $K_d = 0.09 \pm 0.01\,\mu M$ (Fig. 3h), similar to the optimal affinity reported for an interaction between PLK1$_{PBD}$ and its phosphorylated target[23]. Consistently, PLK1$_{PBD}$ bound to the fragment BRCA2$_{190-284}$ with nanomolar affinity upon phosphorylation by PLK1 ($K_d = 0.14 \pm 0.02\,\mu M$; Supplementary Fig. 3a), whereas it did not bind to the corresponding non-phosphorylated polypeptides (Fig. 3i, Supplementary Fig. 3b). Mutation T207A also abolished the interaction (Fig. 3j), in agreement with the pull-down experiments (Fig. 3a–d). More surprisingly, the phosphomimetic substitution T207D was not sufficient to create a binding site for PLK$_{PDB}$ (Supplementary Fig. 2c). A peptide comprising pT207 and the mutation S206C could not bind to PLK1$_{PBD}$ (Fig. 3k), as predicted from the consensus sequence requirement for PLK1$_{PBD}$ interaction[23]. Last, a peptide containing pS197, a predicted docking site for PLK1, bound with much less affinity to PLK1$_{PBD}$ than pT207 ($K_d = 17 \pm 2\,\mu M$; Supplementary Fig. 2d).

Finally, we determined the crystal structure of PLK1$_{PBD}$ bound to the T207 phosphorylated peptide at 3.1 Å resolution (Supplementary Table 2). Analysis of this 3D structure showed that, as expected, the 17 aa BRCA2 phosphopeptide binds in the cleft formed between the two Polo boxes (Fig. 3l). Twelve residues of the peptide (from A199 to I210) are well structured upon binding, burying about 694 Å$^2$ in the interface with PLK1$_{PBD}$. The interface between BRCA2 and PLK1$_{PBD}$ is stabilized by 12 hydrogen bonds: the backbone of residues T200 to L209 as well as the side chain of S206 are bonded to residues from Polo Box 1, whereas the side chain of phosphorylated T207 is bonded to residues from Polo Box 2 (see the zoom view in Fig. 3l). The side chain of S206 participates in two hydrogen-bonding interactions with the backbone of W414, which explains the strict requirement for this amino acid at this position[23]. Moreover, the phosphate group of pT207 participates in three hydrogen-bonding interactions with the side chains of residues H538, K540, and R557 in Polo Box 2 (see the zoom view in Fig. 3l). This explains the critical dependence on phosphorylation for binding observed by ITC (Fig. 3h, i).

Thus, our biochemical and structural analyses demonstrate that the BRCA2-T207 phosphopeptide interacts with PLK1$_{PBD}$ as an optimal and specific PLK1$_{PBD}$ ligand. It supports a mechanism in which phosphorylation of T207 by PLK1 promotes the interaction of PLK1 with BRCA2 through a bona fide docking site for PLK1 and favors a cascade of phosphorylation events. In variant T207A, the absence of T207 phosphorylation impairs

PLK1 docking explaining the reduction of binding to PLK1 and the global loss of phosphorylation by PLK1. S206C eliminates the serine residue at −1 position required for PLK1$_{PBD}$ interaction resulting as well in a strong reduction of BRCA2 binding.

**Impairing T207 phosphorylation prolongs mitosis.** PLK1 is a master regulator of mitosis[17]. To find out whether the interaction between BRCA2 and PLK1 was involved in the control of mitotic progression we examined the functional impact of the variants that reduce PLK1 phosphorylation at T207 (S206C and T207A) in the context of the full-length BRCA2 protein in cells. We generated stable cell lines expressing the BRCA2 cDNA coding for either the GFPMBP-BRCA2 WT or the variants to complement DLD1 BRCA2 deficient human cells (hereafter BRCA2$^{-/-}$) (Supplementary Fig. 4a). First, to confirm that the phosphorylation of T207 by PLK1 takes place in cells we raised a phospho-specific polyclonal antibody against a peptide encompassing pT207. Using an antibody against BRCA2 we detected a band that corresponds to the size of BRCA2 in cell extracts from BRCA2 WT cells, both in mitotic (nocodazole treated) and asynchronous cells, by Western blotting (Fig. 4a). When the same membrane was re-probed with the pT207-BRCA2 antibody a band corresponding to BRCA2 was detected only in the mitotic cells and not in the asynchronous cells (Fig. 4a, lane 1 and 2). In addition, mitotic cell extracts treated with phosphatase lost the signal indicating that the band corresponds to a phosphorylation event (Fig. 4a, lane 3). Finally, cells bearing BRCA2-T207A were detected with the BRCA2 antibody but showed reduced signal for the pT207 antibody (Fig. 4a, lane 4) providing evidence of the specificity of the antibody. We then tested the interaction of full-length BRCA2 with PLK1 in these stable clones by GFP pull-down assay. As expected, PLK1 readily co-purified with full-length BRCA2 WT from mitotic cells. Importantly, in cells expressing the variants S206C and T207A, the level of co-purified PLK1 was greatly reduced (Fig. 4ba, c), confirming the results obtained with the overexpressed BRCA2$_{1-250}$ fragments (Fig. 3a) now in the context of cells stably expressing the full-length BRCA2 protein. Thus, BRCA2 interaction with endogenous PLK1 is impaired in cells bearing variants S206C and T207A.

Also, the binding of BRCA2 to PLK1 in cells stably expressing full-length BRCA2 WT was not reduced by incubating the cells with a PLK inhibitor (BTO) (Supplementary Fig. 4b), consistently with results obtained in vitro (Fig. 3c). Altogether, these data

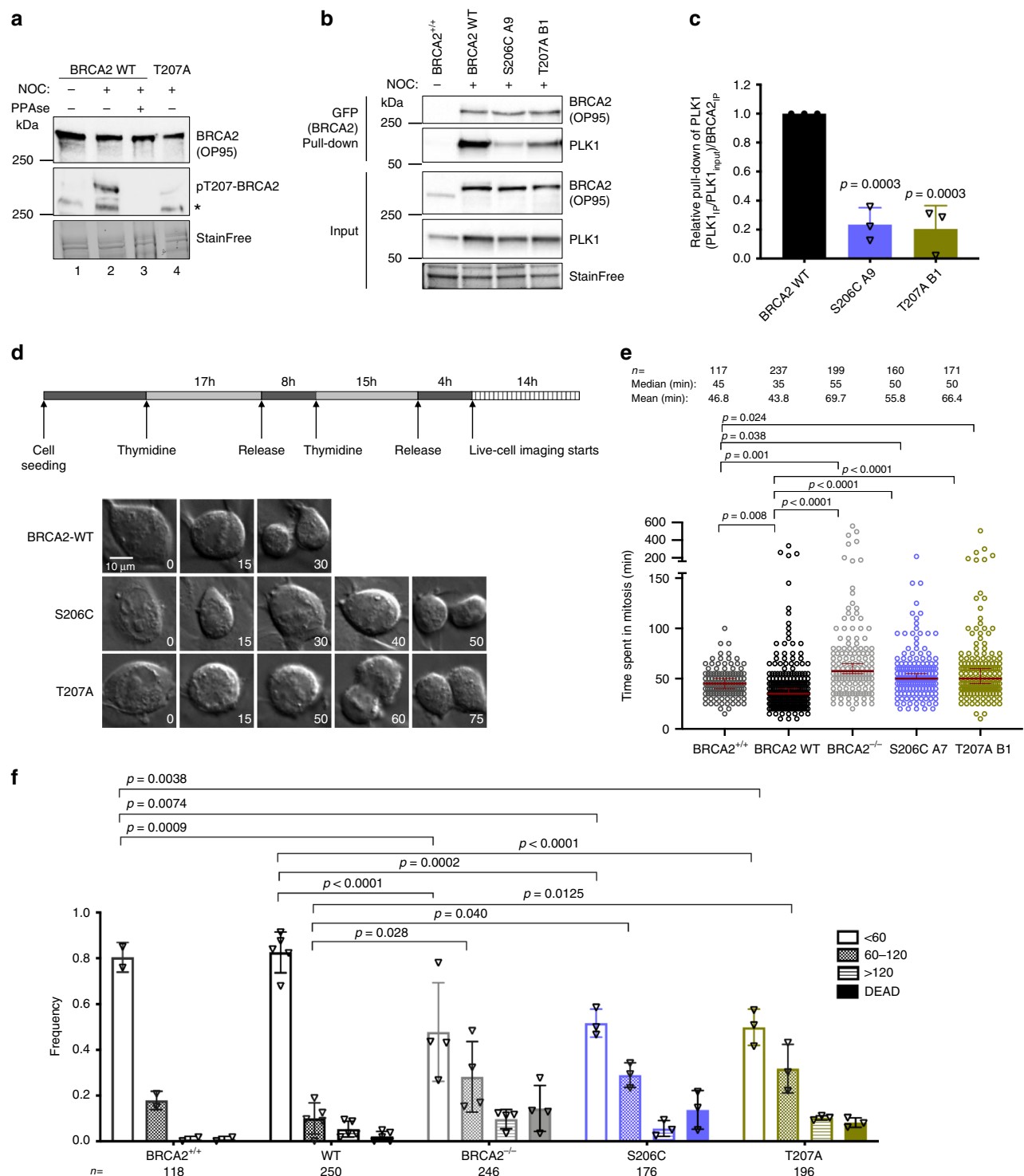

suggest that another binding site primed by a different kinase (presumably T77 phosphorylated by CDK1[29]) contributes to BRCA2 binding to PLK1. Consistent with this idea, the binding of overexpressed 2xMBP-BRCA2$_{1-250}$ to the endogenous PLK1 in U2OS cells was completely abolished in the presence of the CDK inhibitor (RO3306) (Supplementary Fig. 4c, lane 4 compared to lane 2).

Having confirmed that T207 was a docking site for PLK1 in cells, we next examined the impact of BRCA2 variants on mitosis. Therefore, we monitored the time taken for individual cells from

mitotic entry (defined as nuclear envelope break down) to mitotic exit using live cell imaging. Cells expressing the endogenous BRCA2 (hereafter BRCA2$^{+/+}$) and the BRCA2 WT cells showed similar kinetics, they completed mitosis, on average, in 47 and 44 min, respectively (Fig. 4d, e) and the majority of the cells (80% for BRCA2$^{+/+}$ and 82% for BRCA2 WT) completed mitosis within 60 min (Fig. 4f). In contrast, cells expressing variants S206C and T207A augmented the time spent in mitosis (average time of 56 and 66 min, respectively, Fig. 4e). This trend was also observed in the frequency of cells dividing within 60 min (~49–51%),

**Fig. 4 Cells bearing BRCA2 variants S206C and T207A prolong mitosis. a** Protein levels of BRCA2 and pT207-BRCA2 in cells bearing BRCA2 WT or the T207A variant from whole cell lysates of nocodazole-arrested (100 ng/µl for 14 h) or asynchronous cells. 4–15% SDS-PAGE followed by WB using anti-BRCA2. The same blot was stripped and re-probed with anti-pT207-BRCA2 antibody. Lane 3 protein extracts were pre-treated with phosphatase (FastAP) for 1 h before loading onto the gel. Asterisk indicates a non-specific band. (**b**) GFP-trap pull-down of EGFPMBP-BRCA2 from cells bearing BRCA2 WT, S206C or T207A. 4–15% SDS-PAGE followed by WB using anti-BRCA2 and -PLK1 antibodies. Asynchronous DLD1 cells with endogenous BRCA2 (BRCA2$^{+/+}$) were used as control for the pull-down and StainFree images of the gels before transfer as loading control (cropped image is shown). **c** Quantification of co-immunoprecipitated PLK1 with EGFP-MBP-BRCA2 in (**b**) relative to the PLK1 protein levels in the input and the amount of pull-down EGFP-MBP-BRCA2 ((PLK1$_{IP}$ /PLK1$_{input}$)/EGFP-MBP-BRCA2$_{IP}$) Results are presented as the fold change compared to the BRCA2 WT clone. The data represent the mean ± SD of three independent experiments. One-way ANOVA test with Dunnett's multiple comparisons test was used to calculate statistical significance of differences (p-values compared to WT). **d** Top: Synchronization scheme. Bottom: Representative still images of the time-lapse videos. Numbers represent time (min) after nuclear envelope break down (NEBD). Scale bar represents 10 µm. **e** Quantification of the time the cells spent in mitosis in (**d**). The red line indicates the median (95% CI). Each dot represents a single cell, n is the total number of cells from two to four independent experiments (45–60 cells per experiment) (BRCA2$^{+/+}$ (n = 2), WT C1 (n = 5), BRCA2$^{-/-}$ (n = 4), S206C A7 (n = 3), T207A B1 (n = 3)). Kruskal–Wallis one-way analysis followed by Dunn's multiple comparison test was used to calculate statistical significance of differences. **f** Frequency distribution of the time spent in mitosis in (**d**), including cells that fail to divide (DEAD). The error bars represent mean ± SD of two to four independent experiments (BRCA2$^{+/+}$ (n = 2), WT C1 (n = 5), BRCA2$^{-/-}$ (n = 4), S206C A7 (n = 3), T207A B1 (n = 3)). Two-way ANOVA test with Tukey's multiple comparisons test was used to calculate statistical significance of differences. Source data are available as a Source Data file.

compared to 82% in BRCA2 WT cells (Fig. 4f). Representative videos of the still images shown in Fig. 4d are included in Supplementary movies 1–3.

Taken together, the phosphorylation of T207 takes place in cells in mitosis. Cells altering this phosphorylation (bearing S206C and T207A variants) display a significant delay in mitotic progression compared to BRCA2 WT cells.

**Docking of PLK1 at T207-BRCA2 favors a complex with PP2A and pBUBR1.** BRCA2 forms a complex with BUBR1[4,5]. BUBR1 facilitates kinetochore–microtubule attachments via its interaction with the phosphatase PP2A. Phosphorylation of BUBR1 by PLK1 at the KARD motif comprising the tension-sensitive sites S676 and T680 promotes interaction with PP2A[19]. A defect in the phosphorylation of BUBR1 weakens its interaction with PP2A leading to mitotic delay[20,30]. The mitotic delay phenotype we observed in BRCA2 mutated cell lines led us to ask whether BRCA2 and PLK1 formed a tetrameric complex with pBUBR1 and PP2A. Using a GFP pull-down to capture GFP-MBP-BRCA2 from mitotic BRCA2 WT cells, we observed that PLK1, pT680-BUBR1 and PP2A were pull-down together with GFP-MBP-BRCA2, indicating the formation of a tetrameric complex (Fig. 5a). As described for pBUBR1[18,19], PLK1[31] and PP2A[21], we found BRCA2 at the kinetochore in mitotic cells (Supplementary Fig. 5a) as previously reported[4] supporting the idea that this complex takes place at the kinetochore.

Importantly, cells expressing the variants S206C or T207A showed a strong reduction in the interaction of BRCA2 with PLK1, PP2A, BUBR1 and pT680-BUBR1 in the context of the tetrameric complex (Fig. 5a, b). Moreover, the overall levels of BUBR1 and pBUBR1 were also reduced in cells bearing S206C and T207A variants compared to the WT cells, as detected by specific antibodies against BUBR1, pT680-BUBR1 (Fig. 5e–i) and pS676-BUBR1 (Supplementary Fig. 5d), and this was also the case in BRCA2 deficient cells (DLD1 BRCA2$^{-/-}$ cells) or U2OS cells depleted of BRCA2 by siRNA (Fig. 5f, g). Furthermore, we observed an overall reduction in the levels of pBUBR1 at the kinetochore (Fig. 5h, i) in cells expressing T207A compared to WT cells. Consistently, when we immunoprecipitated BUBR1 from mitotic cells and detected the levels of co-immunoprecipitated PP2A (PP2AC antibody), we observed that, although PP2A was readily copurified with BUBR1 in the BRCA2 WT cells, expressing BRCA2 variant T207A reduced the levels of PP2A by ~30% (Fig. 5c, d) suggesting that BRCA2 facilitates the formation of a complex between BUBR1-and PP2A.

**Impaired phosphorylation of T207 leads to chromosome misalignment.** The association of PLK1-phosphorylated BUBR1 with PP2A is required for the formation of stable kinetochore–microtubule attachments[18,19], a defect in this interaction resulting in chromosome misalignment. Therefore, we next examined whether cells expressing the BRCA2 variants S206C and T207A altered chromosome alignment. Following thymidine synchronization, the cells were treated with the Eg5 inhibitor Monastrol (100 µM) for 14 h followed by Monastrol washout and release for 1 h in normal media supplemented with the proteasome inhibitor MG132 to avoid exit from mitosis[18]. Chromosome alignment was then analyzed by immunofluorescence. Importantly, the analysis of cells expressing S206C and T207A variants showed high frequency of faulty chromosome congression compared to the BRCA2 WT clone (47% in S206C and 38% in T207 versus 24% in the BRCA2 WT clone), which was exacerbated in BRCA2$^{-/-}$ cells (63%) (Fig. 6a, b), as detected by signals of the centromere marker (CREST) outside the metaphase plate (Fig. 6b). Next, to find out if this defect in alignment was due to impaired stability of kinetochore–microtubule interactions as previously reported[6,18], we examined the presence of cold stable microtubules in cells bearing T207A mutation compared to BRCA2 WT cells. Following the same synchronization procedure as before (Fig. 6a), the cells were kept on ice for 15 min before fixation. Under these conditions, BRCA2 WT metaphase cells exhibited relatively intact bipolar spindles with most CREST-stained kinetochores attached to the microtubules (α-tubulin) (Fig. 6c). In stark contrast, almost all the kinetochore–microtubules attachments were lost in T207A mutated cells upon cold treatment (Fig. 6c) as measured by the relative intensity of α-tubulin in these cells with respect to BRCA2 WT cells (~6-fold reduction of the median) (Fig. 6d).

To better understand the mechanism behind the phenotype observed in our clones, we overexpressed a phosphomimic version of BUBR1 RFP-BUBR1-3D (S670D, S676D, T680D) in T207A mutated cells to test whether this form of BUBR1 could restore the misalignment phenotype as previously shown for BUBR1 depleted cells[19]. Surprisingly, T207A bearing cells overexpressing RFP-BUBR1-3D exhibited a similar misalignment phenotype (~38% of cells with misalignment) compared to the non-transfected cells (33%) (Fig. 7a, b). Moreover, BUBR1-3D, previously shown to be sufficient to restore PP2A binding in BUBR1 deficient cells[19], could not rescue the interaction of BUBR1 with PP2A in cells bearing BRCA2 S206C variant (Fig. 7c). This effect was not due to an increased localization of PLK1 at the kinetochore in the mutated cells, as the levels of PLK1 at the kinetochores remained unchanged (Supplementary

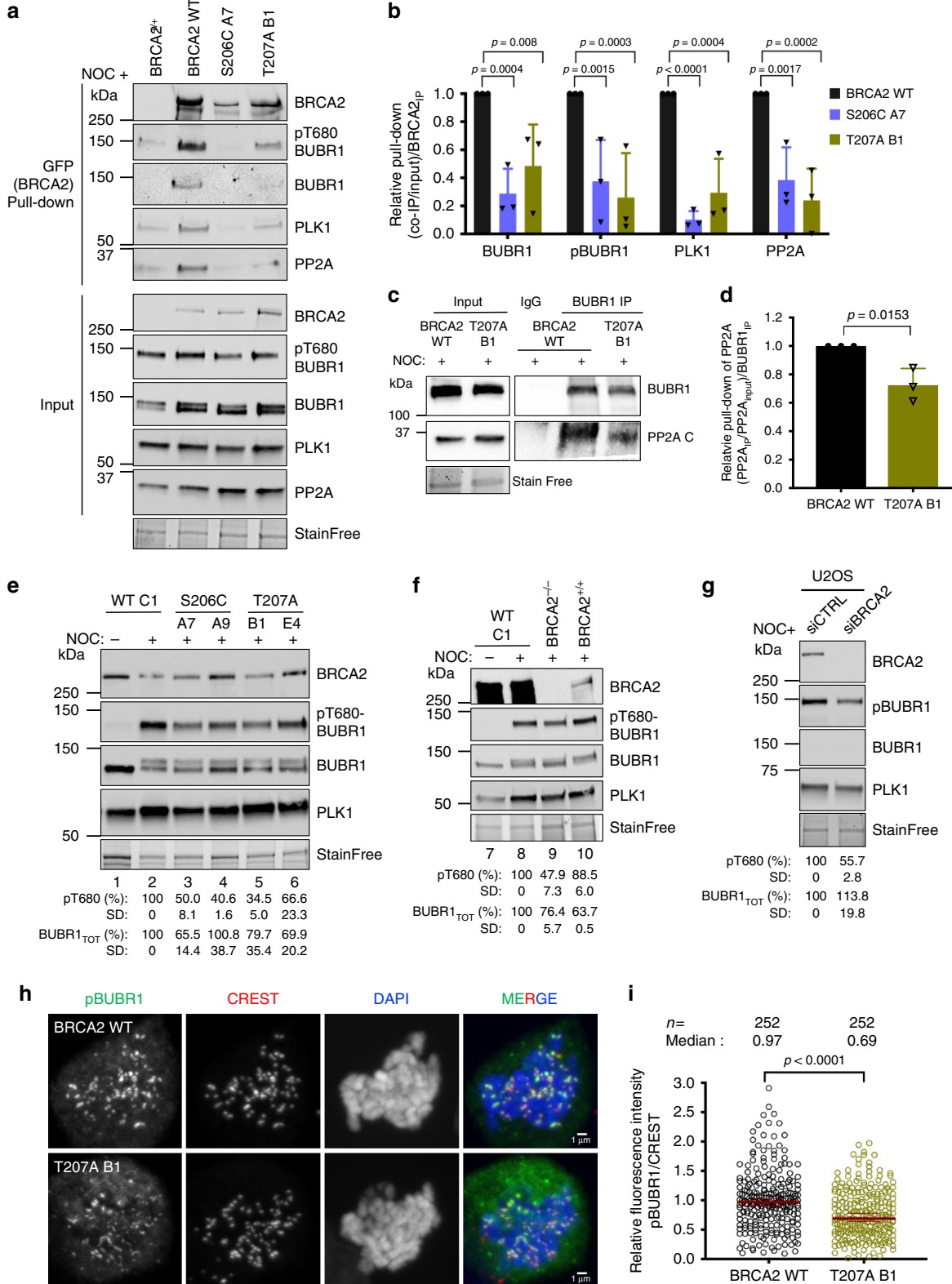

Fig. 6a, b), nor to an increased interaction of PLK1 with BUBR1 in the mutated cells (Supplementary Fig. 6c, d).

Together, these results strongly suggest that docking of PLK1 onto BRCA2 T207 facilitates the formation of a complex between phosphorylated BUBR1 and PP2A at the kinetochore that is essential for the stability of microtubule–kinetochore attachments with direct consequences in chromosome alignment. The fact that BUBR1-3D cannot rescue the phenotype

favors the hypothesis that BRCA2 is required for PP2A-BUBR1 interaction.

**Reduced T207 phosphorylation lead to defects in chromosome segregation.** Unresolved chromosome misalignment as observed in cells altering BRCA2 phosphorylation by PLK1 is expected to drive chromosome missegregation. To find out if this was the case

**Fig. 5 S206C and T207A impair the complex of BRCA2 with PLK1-BUBR1-PP2A and reduce the levels of pBUBR1 at the kinetochore. a** Pull-down of BRCA2 using GFP-trap beads from mitotic cell extracts of cells bearing BRCA2 WT cells or the variant S206C and T207A. Complexes containing BRCA2-BUBR1/pBUBR1-PP2A-PLK1 were detected by 4–15% SDS-PAGE followed by WB using anti-BRCA2, -BUBR1, -pT680-BUBR1, -PLK1 and -PP2AC (PP2A catalytic subunit) antibodies. Mitotic BRCA2$^{+/+}$ cells were used as control pull-down. **b** Quantification of co-immunoprecipitated BUBR1, pBUBR1, PLK1 and PP2A with EGFPMBP-BRCA2 in (**a**), relative to the input levels of each protein and the amount of pull-down EGFP-MBP-BRCA2. Results are presented as the fold change compared to the BRCA2 WT clone. The data represent the mean ± SD of three independent experiments. Two-way ANOVA test with Dunnett's multiple comparisons test was used to calculate statistical significance of differences. **c** IP of endogenous BUBR1 from mitotic cell extracts of BRCA2 WT cells or BRCA2-T207A using mouse anti-BUBR1 antibody. Mouse IgG was used as control. 4–15% SDS-PAGE followed by WB using rabbit anti-BUBR1 and anti-mouse PP2AC antibodies. **d** Quantification of co-IPed PP2A in (**c**), relative to the input levels and the amount of IPed BUBR1. Results are presented as the fold change compared to the BRCA2 WT clone. The data represent the mean ± SD of three independent experiments. Unpaired two-tailed t-test was used to calculate statistical significance of differences. **e–g** WB showing the protein levels of endogenous BUBR1 and pT680-BUBR1 in nocodazole treated cells bearing BRCA2 WT or the variants, as indicated (**e**) BRCA2$^{-/-}$ or BRCA2$^{+/+}$ (**f**). **g** WB showing the protein levels of endogenous BUBR1 and pT680-BUBR1 in U2OS after depletion of endogenous BRCA2 by siRNA. **e–g** The mean BUBR1$_{TOT}$ and pBUBR1 signal relative to the stain free signal is shown for the nocodazole treated samples below the blots, results are presented as percentage compared to BRCA2 WT clone. The data represent the mean ± SD of three (**e**) and two (**f** and **g**) independent experiments. The protein levels of PLK1 in (**a**, **e–g**) are shown as a G2/M marker. **h** Representative images of the localization of pT680-BUBR1 in cells bearing BRCA2 WT or the variant T207A as indicated. CREST is used as centromere marker and DNA is counterstained with DAPI. Scale bar represents 1 μm. **i** Quantification of the co-localization of pT680-BUBR1 and CREST in (**h**). The data represent the intensity ratio (pT680-BUBR1:CREST) relative to the mean ratio of pT680-BUBR1:CREST for the GFP-MBP-BRCA2 WT calculated from two independent experiments (252 pairs of chromosomes analyzed). The red line in the plot indicates the median (95% CI) ratio, each dot represents a pair of chromosomes. Mann–Whitney two-tailed analysis was used to calculate statistical significance of differences. Source data are available as a Source Data file.

in cells expressing BRCA2 variants S206C and T207A, we examined chromosome segregation by immunofluorescence in cells synchronized by double-thymidine block and released for 15 h to enrich the cell population at anaphase/telophase stage (Fig. 7d). BRCA2$^{-/-}$ cells displayed, as expected, an increased proportion of chromosome bridges (39% vs 16% in cells expressing BRCA2 WT), whereas the fraction of lagging chromosomes was only mildly increased (7% vs 4% in BRCA2 WT). The percentage of chromosome bridges in cells expressing S206C and T207A was moderately increased (23% and 29%, respectively, compared to 16% in BRCA2 WT). However, the biggest difference was observed in the percentage of lagging chromosomes increasing between 3- and 5-fold in the cells bearing the variants compared to the BRCA2 WT cells (Fig. 7d, e).

Erroneous chromosome segregation generates aneuploid cells during cell division[30]. Given the strong chromosome segregation defects observed in cells expressing S206C and T207A we next analyzed the number of chromosomes in these cells. Total chromosome counts carried out on metaphase spreads revealed that 37.1% of BRCA2 WT cells exhibited aneuploidy with chromosome losses or gains. In the case of S206C and T207A, this number was elevated to 52.2% and 61.8% of the cells, respectively (Fig. 8a). An example of the images analyzed can be found in Fig. 8b. As the number of chromosomes was difficult to assess for cells with high content of chromosome gains we arbitrarily discarded cells that contained more than 65 chromosomes. Thus, tetraploid cells were not included in this measurement. Therefore, we determined the frequency of tetraploid cells by assessing the incorporation of BrdU and measuring the frequency of S-phase cells with >4 N DNA content (Fig. 8c). The frequency of tetraploidy in cells bearing the variants was <1% of the total population as in the BRCA2 WT cells (Fig. 8d), and the number of BrdU positive cells was also equivalent (Supplementary Fig. 7).

Together, these results indicate that, in addition to the severe chromosome misalignment phenotype, cells expressing S206C and T207A display high frequency of chromosome missegregation, including a strong induction of lagging chromosomes and a mild increase in chromosome bridges. As a consequence, the incidence of aneuploidy, but not tetraploidy, is greatly exacerbated in these cells.

**The variants reducing PLK1 phosphorylation of BRCA2 do not alter HR.** Since BRCA2 has a major role in DNA repair by HR,

the prolonged mitosis observed in the VUS-expressing stable cell lines (Fig. 4) could result from checkpoint activation through unrepaired DNA. Thus, we performed a clonogenic survival assay in the stable clones after treatment with mitomycin C (MMC), an inter-strand crosslinking agent to which BRCA2 deficient cells are highly sensitive[32–35]. As expected, BRCA2 deficient cells (BRCA2$^{-/-}$) showed hypersensitivity to MMC treatment whereas BRCA2 WT cells complemented this phenotype almost to the same levels as the cells expressing the endogenous BRCA2 (BRCA2$^{+/+}$). Cells bearing variants S206C and T207A also complemented the hypersensitive phenotype of BRCA2$^{-/-}$ cells, although there was a mild effect compared to the BRCA2 WT cells (Fig. 9a). These results suggest that the delay in mitosis is not a consequence of checkpoint activation via unrepaired DNA.

Cells expressing VUS S206C and T207A showed a growth defect manifested in a reduced number of colonies (Supplementary Fig. 8a). To exclude a possible bias arising from the different ability to form colonies we used MTT assay. As shown in Fig. 9b, cells expressing S206C and T207A showed similar relative viability upon MMC treatment compared to BRCA2 WT cells or BRCA2$^{+/+}$, confirming our results. Similarly, the viability upon treatment with the poly (ADP-ribose) polymerase inhibitor (PARPi) Olaparib was not affected in cells bearing the variants (Fig. 9c).

To determine directly the levels of spontaneous DNA damage in these cells and their ability to form DNA repair foci, we measured the number of nuclear foci of the DSB marker γH2AX and RAD51 protein, in cells unchallenged (−IR) or 2 h after exposure to ionizing radiation (6 Gy, (+IR)). Our results show that in the absence of DNA damage, the number of γH2AX foci or RAD51 foci is comparable in all cell lines including in cells depleted of BRCA2 (Fig. 9d and Supplementary Fig. 8b); this is probably due to the high genome instability intrinsic to these cancer cells. In contrast, the number of RAD51 foci upon irradiation increased 5-fold in BRCA2$^{+/+}$ and BRCA2 WT cells and 3-fold in T207A bearing cells while it remained low in BRCA2 depleted cells, as expected (Fig. 9e). We conclude that the DNA repair foci are only mildly altered in cells expressing T207A. Representative images of these experiments are shown in Supplementary Fig. 8b.

A typical feature of replication stress is the appearance of micronuclei in daughter cells which generally contain DNA fragments. In contrast, micronuclei with centromeres suggest an event arising from lagging chromosomes involving whole chromosomes or chromatids. BRCA2 deficient cells displayed

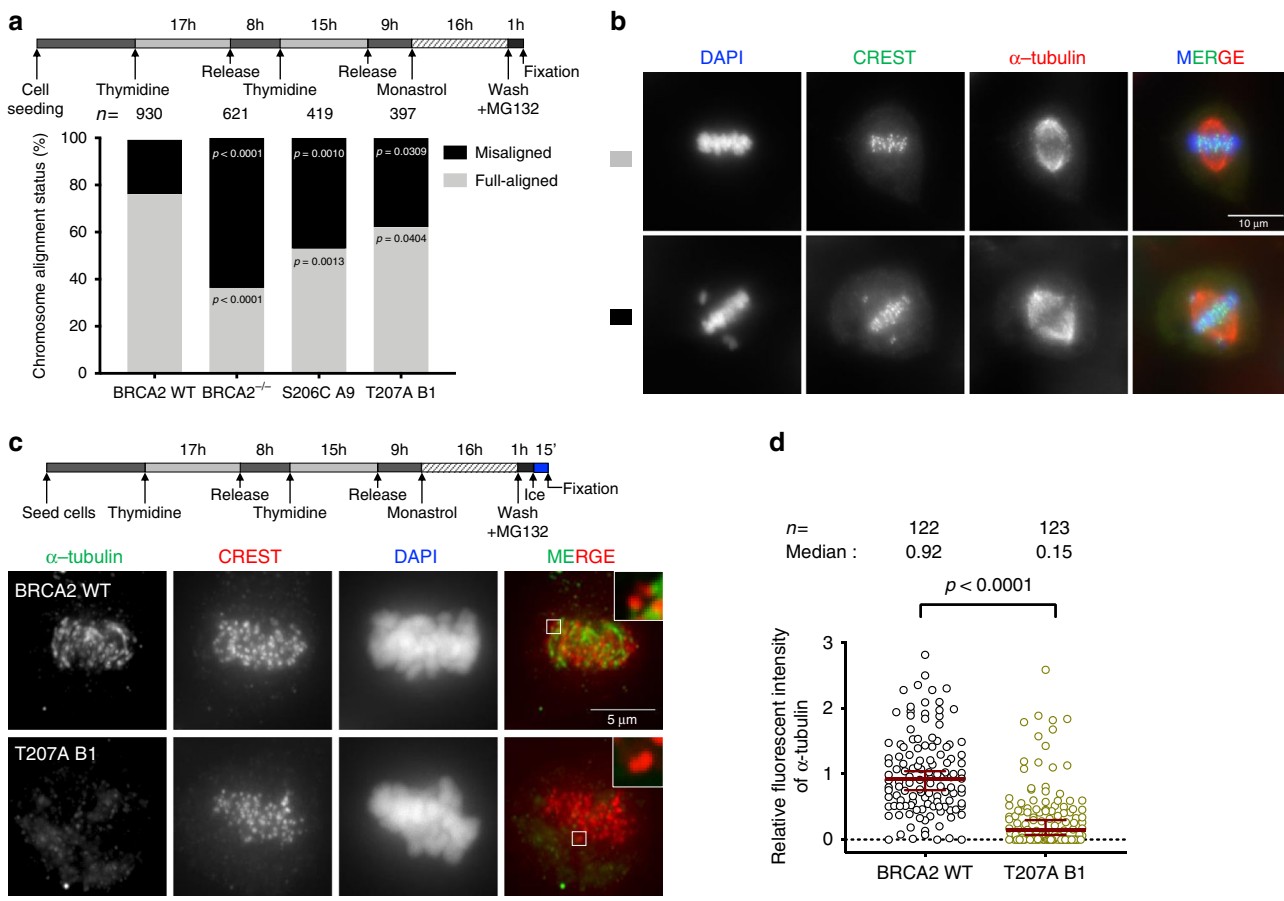

**Fig. 6 Cells expressing BRCA2 variants S206C and T207A display reduced stability of kinetochore–microtubule attachments and misaligned chromosomes. a** Top: Scheme of the double thymidine block procedure used to synchronize the DLD1 cells for the analysis of chromosome alignment. Bottom: Quantification of misaligned chromosomes outside the metaphase plate in DLD1 BRCA2 deficient cells (BRCA2$^{-/-}$) and BRCA2$^{-/-}$ cells stably expressing BRCA2 WT or the S206C and T207A variants. *n* indicates the total number of cells counted for each clone from two (BRCA2$^{-/-}$, S206C, and T207A) and four (BRCA2 WT) independent experiments. Statistical significance of the difference was calculated with unpaired two-way ANOVA test with Tukey's multiple comparisons test, the p-values show the significant differences. **b** Representative images of the type of chromosome alignment observed in cells quantified in (**a**), scale bar represents 10 µm. **c** Top: Scheme of the synchronization procedure for the analysis of cold stable microtubules in the BRCA2-WT and T207A stable clones. Bottom: Representative images of cold stable microtubules in cells expressing BRCA2 WT or the T207A variant. Cells treated according to the scheme were co-stained with α-tubulin and CREST, as centromere marker and DNA was counterstained with DAPI. Scale bar represents 5 µm. A zoom-in inset in the images show representative kinetochore–microtubule attachments in each cell line. **d** Quantification of the relative intensity of α-tubulin normalized to the one in BRCA2 WT cells in (**c**). The data represent the results from two independent experiments from a total of 122 (WT) and 123 (T207A) cells. The red line in the plot indicates the median intensity (95% CI), each dot representing the intensity for one cell. For statistical comparison of the differences between the samples we applied Mann–Whitney two-tailed analysis, the p-values show significant differences. Source data are available as a Source Data file.

an increased number of both types of micronuclei compared to BRCA2 WT cells. In contrast, cells bearing S206C or T207A variant did not change the number of micronuclei (Supplementary Fig. 8d, e) excluding strong replication stress-induced DNA damage in these cells.

Finally, to directly assess the HR proficiency, we performed a cell-based HR assay by DSB-mediated gene targeting using a site-specific transcription-activator like effector nuclease (TALEN) and a promoter-less mCherry donor flanked by homology sequence to the targeted locus[36]. DSB-meditated gene targeting results in mCherry expression from the endogenous promoter (Fig. 9f) which can be measured by flow cytometry (Supplementary Fig. 9). Using this system, BRCA2$^{+/+}$ and BRCA2 WT cells showed ~5% of mCherry positive TALEN-transfected cells (mean of 5.6% for BRCA2$^{+/+}$ and 4.9% for WT) whereas BRCA2$^{-/-}$ exhibited reduced mCherry expressing cells (1.8%), as expected. Importantly, TALEN-transfected cells expressing BRCA2 variants S206C and T207A showed no significant difference with the BRCA2 WT.

In summary, these results indicate that the role of BRCA2 in conjunction with PLK1 in mitosis is likely independent of the HR function of BRCA2 as the variants S206C and T207A affecting PLK1 phosphorylation of BRCA2 are only mildly sensitive to DNA damage, do not show an increased number of micronuclei, are able to recruit RAD51 to DNA damage sites (as shown for T207A) and are efficient at DSB-mediated gene targeting.

## Discussion

Our results demonstrate that residues S193 and T207 of BRCA2 can be efficiently phosphorylated by PLK1 (Fig. 2), thus extending the consensus sequence for phosphorylation by this kinase: position 205 is a serine and not a negatively charged residue, as generally observed at PLK1 phosphorylation sites[23]. Moreover, pT207 constitutes a bona fide docking site for PLK1$_{PBD}$ (Fig. 3h–l) that is phosphorylated in mitotic cells (Fig. 4a). Accordingly, *BRCA2* missense VUS reducing the phosphorylation

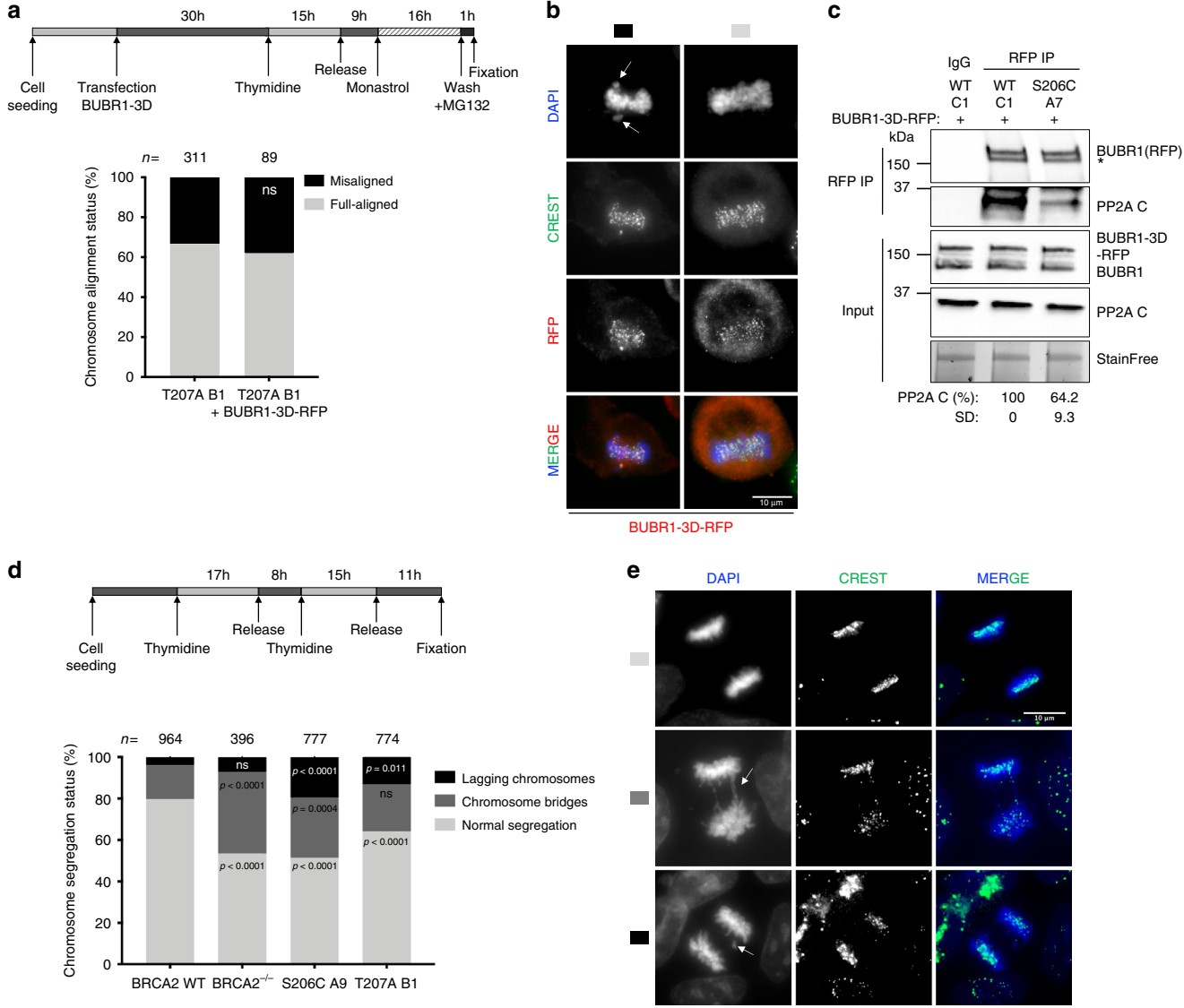

**Fig. 7 Cells expressing BRCA2 variants S206C and T207A display aberrant chromosome segregation. a** Top: Scheme of the synchronization procedure. Bottom: Quantification of misaligned chromosomes in BRCA2-T207A stable clones transiently overexpressing the 3xFLAG-BUBR1-3D-RFP mutant (S670D, S676D and T680D). $n$ indicates the total number of cells counted from two independent experiments. Unpaired two-tailed $t$-test was used to calculate statistical significance of differences, ns (non-significant)). **b** Representative images of the type of chromosome alignment observed in cells overexpressing 3xFLAG-BUBR1-3D-RFP quantified in (**a**); scale bar represents 10 μm. **c** Co-immunoprecipitation of endogenous PP2A with transiently overexpressed 3xFLAG-BUBR1-3D-RFP mutant from mitotic cell extracts of BRCA2 WT cells or cells bearing the S206C variant using rabbit anti-RFP antibody. Rabbit IgG was used as control for the BUBR1 immunoprecipitation. The immunocomplexes were resolved on 4–15% SDS-PAGE followed by WB using mouse anti-BUBR1 and PP2AC antibodies. The amount of PP2A co-immunoprecipitated with BUBR1-3D-RFP relative to the input levels of PP2A and the amount of immunoprecipitated BUBR1-3D-RFP is presented below the blot as mean ± SD from two independent experiments. The data are presented relative to the non-treated BRCA2 WT. Asterisk denotes a non-specific band (**d**). Top: Scheme of the synchronization procedure. Bottom: Quantification of cells with aberrant chromosomes segregation in BRCA2$^{-/-}$ cells and in the clones stably expressing BRCA2 WT, S206C and T207A, as indicated. Two-way ANOVA test with Tukey's multiple comparisons test was used to calculate statistical significance of differences (the $p$-values show the significant differences compared to WT; ns (non-significant)). $n$ in (**d**) indicates the total number of cells counted for each clone from two (BRCA2$^{-/-}$, S206C and T207A) and five (BRCA2 WT) independent experiments. **e** Representative images of the type of aberrant chromosome segregation observed in the cells quantified in (**d**), CREST antibody is used as marker of centromere; nuclei are revealed with DAPI counterstaining. Scale bar represents 10 μm. Source data are available as a Source Data file.

status of T207 (T207A, S206C) result in a decrease in BRCA2-PLK1 interaction (Fig. 3a–k, 4b). Cells expressing these two breast cancer variants in a BRCA2 deficient background display defective chromosome congression (Fig. 6a, b) to the metaphase plate due to a reduced microtubule–kinetochore stability (Fig. 6c, d), causing a substantial delay in mitosis progression (Fig. 4d–f).

Proper kinetochore–microtubule attachments require the interaction of BUBR1 with the phosphatase PP2A-B56 to balance

Aurora B kinase activity[19,21]. This interaction is mediated through the phosphorylation of the KARD motif of BUBR1 by PLK1. BRCA2 does not alter PLK1 interaction with BUBR1 (Supplementary Fig. 6c, d). However, we found that BRCA2 forms a tetrameric complex with PLK1-pBUBR1-PP2A, and that this complex is strongly reduced in cells bearing BRCA2 variants S206C and T207A (Fig. 5a, b). Furthermore, cells bearing BRCA2 variants S206C and T207A show reduced overall levels of BUBR1

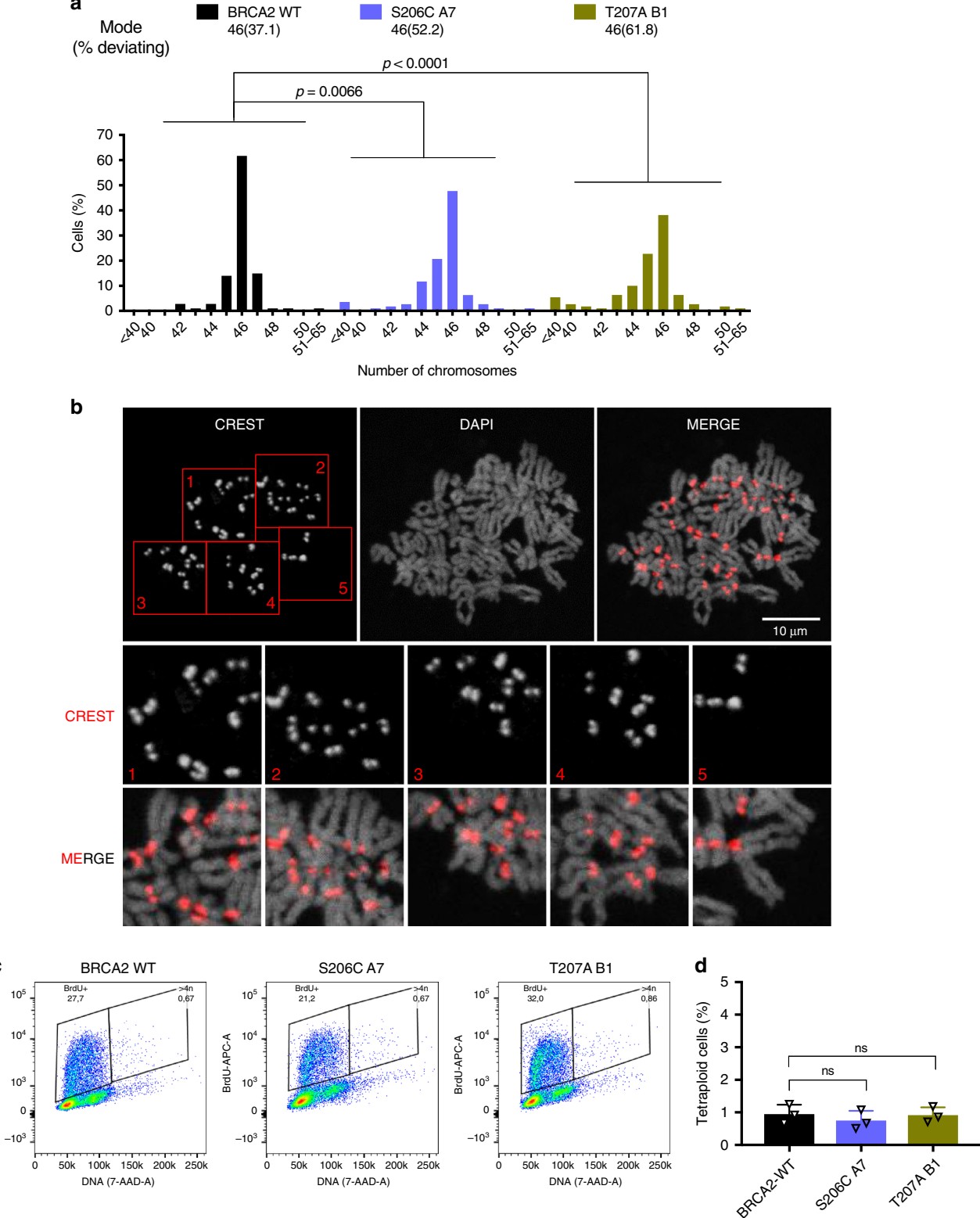

including pBUBR1(pT680) at the kinetochore (Fig. 5i). Importantly, the fact that in the BRCA2 mutated cells the interaction of PP2A with total BUBR1 is reduced (Fig. 5c, d) and that neither this interaction (Fig. 7c) nor the chromosome alignment defect can be rescued by BUBR1-3D overexpression (Fig. 7a, b) strongly suggests that PP2A needs to be in complex with PLK1-bound BRCA2 to bind BUBR1 and facilitate stable kinetochore–microtubules attachment for proper chromosome alignment.

Cells bearing T207A and S206C variants display chromosome segregation errors including lagging chromosomes and chromosome bridges (Fig. 7d, e). Importantly, these accumulated errors ultimately led to a broad spectrum of chromosome gains and losses (aneuploidy) compared to the wild type counterpart (Fig. 8a), but not to tetraploid cells (Fig. 8c, d), suggesting that cytokinesis per se, in which BRCA2 is also involved[11–13], is not affected.

**Fig. 8 Cells expressing BRCA2 variants S206C and T207A exhibit aneuploidy. a** Distribution of the number of chromosomes observed in metaphase spreads of stable clones expressing BRCA2 WT, S206C A7 or T207 B1, total number of cells counted; BRCA2 WT ($n = 105$), S206C A7 ($n = 111$) and T207A B1 ($n = 110$) from two independent experiments. Modal number of chromosomes and percentage deviating from the mode are shown at the top. Kruskal-Wallis one-way analysis followed by Dunn's multiple comparison test was used to calculate statistical significance of differences. The cell passage was between 5 and 9 (BRCA2 WT (p.6 and p.9), S206C A7 (p.5 and p.9) and T207A B1 (p.6 and p.9). **b** Representative image of two independent experiments of metaphase spreads of the DLD1 BRCA2 deficient stable cells bearing the S206C BRCA2 variant stained with CREST and counterstained with DAPI. In this example, the cell contains 45 chromosomes. **c–d** Analysis of S-phase tetraploid cells in cells bearing BRCA2 WT or the VUS S206C and T207A measured by flow cytometry after 20 min of BrdU incorporation. **c** Representative flow cytometry plots of cells stained with anti-BrdU-APC antibodies and 7-AAD (DNA). **d** Frequency of S-phase tetraploid cells in stable clones expressing BRCA2 WT or the VUS S206C and T207A. The data represents the mean ± SD of three independent experiments (cell passage: 6–10). One-way ANOVA test with Tukey's multiple comparisons test was used to calculate statistical significance of differences (the $p$-values show the difference compared to WT, ns: non-significant). Source data are available as a Source Data file.

Finally, the function of BRCA2-PLK1 interaction in mitosis seems to be independent of the HR function of BRCA2 as cells expressing these variants display mild sensitivity to DNA damage (MMC) and PARP inhibitors (Fig. 9a–c), normal recruitment of RAD51 to DNA damage sites (Fig. 9e, Supplementary Fig. 8b), absence of micronuclei (Supplementary Fig. 8d, e), rescue the chromosome bridges phenotype of BRCA2 deficient cells (Fig. 7d) and their HR activity, as measured by a DSB-mediated gene targeting assay (Fig. 9f, Supplementary Fig. 9). Nevertheless, we cannot rule out that the mild phenotype observed in cells bearing S206C and T207A may arise from a disrupted interaction with an unknown DNA repair factor that would bind to the region of BRCA2 where these variants localize.

Putting our results together we reveal an unexpected chromosome stability control mechanism that depends on the phosphorylation of BRCA2 by PLK1 at T207. We show that BRCA2 pT207 is a docking platform for PLK1 that ensures the efficient interaction of BUBR1 with PP2A phosphatase required for chromosome alignment. We propose the following working model (Fig. 9g): in cells expressing BRCA2 WT, PLK1 phosphorylates BRCA2 on T207 leading to the docking of PLK1 at this site. This step promotes the formation of a complex between BRCA2-PLK1-pBUBR1-PP2A in prometaphase at the kinetochore, leading to an enrichment of phosphorylated BUBR1 and the phosphatase PP2A to balance Aurora B activity (Fig. 9g, panel 1). Why is BRCA2 required for PP2A interaction with pBUBR1 and, is this regulated by PLK1? It has been reported that BRCA2 fragment from aa 1001 to aa 1255 (BRCA2$_{1001-1255}$) comprising a PP2A-B56 binding motif, binds to the B56 subunit of PP2A; in addition, the phosphorylation of positions 2 and 8 of this motif enhances the binding to B56[37]. In BRCA2, these positions are occupied by serines that are targets for PLK1. Therefore, it is likely that the recruitment of PLK1 by BRCA2 T207 would favor phosphorylation of BRCA2$_{1001-1255}$ and binding of the complex between BRCA2 and PLK1 to PP2A-B56. Non-exclusively, BUBR1 has been reported to bind directly BRCA2 although there are inconsistencies regarding the site of interaction[4,5]. Thus, either the direct interaction of BRCA2 with PP2A or with BUBR1 or both could be favoring the formation of a complex between PP2A and pBUBR1.

In cells expressing the variants that impair T207 phosphorylation (S206C, T207A), PLK1 cannot be recruited to pT207-BRCA2, impairing the formation of the complex with PLK1, PP2A and BUBR1 (Fig. 9g, panel 1′), which in turn reduces the amount of pBUBR1 and its binding to PP2A for stable kinetochore–microtubule interactions. This leads to chromosome misalignment defects that prolong mitosis (Fig. 9g, panel 2′); as a consequence, these cells exhibit increased chromosome segregation errors (Fig. 9g, panel 3′) and aneuploidy.

Although the individual BRCA2 variants analyzed here are rare (Supplementary table 1), the majority of pathogenic mutations recorded to date lead to a truncated protein either not expressed

or mislocalized[38] which would be predicted to affect this function. Consistent with this idea, the BRCA2 deficient cells or cells transiently depleted of BRCA2 used in this study exhibit low levels of phosphorylated BUBR1 (Fig. 5f, g). Thus, the chromosome alignment function described here could be responsible, at least in part, for the numerical chromosomal aberrations observed in *BRCA2*-associated tumors[39].

Finally, the lack of sensitivity to the PARP inhibitor Olaparib observed in our cell lines (Fig. 9c) has important clinical implications as breast cancer patients carrying these variants are not predicted to respond to PARP inhibitor treatment (unlike *BRCA2*-mutated tumors that are HR-deficient).

## Methods

**Cell lines, cell culture, and synchronizations.** The human cell lines HEK293T and U2OS cells (kind gift from Dr. Mounira Amor-Gueret) were cultured in DMEM (Eurobio Abcys, Courtaboeuf, France) media containing 25 mM sodium bicarbonate and 2 mM L-Glutamine supplemented with 10% heat inactive FCS (EuroBio Abcys). The BRCA2 deficient colorectal adenocarcinoma cell line DLD1 BRCA2$^{-/-}$ (Hucl, T. et al 2008) (HD 105-007) and the parental cell line DLD1 BRCA2$^{+/+}$ (HD-PAR-008) was purchased from Horizon Discovery (Cambridge, England). In DLD1 BRCA2$^{-/-}$ cell line, both alleles of BRCA2 contain a deletion in exon 11 causing a premature stop codon after BRC5 and cytoplasmic localization of a truncated form of the protein[40]. The cells were cultured in RPMI media containing 25 mM sodium bicarbonate and 2 mM L-Glutamine (EuroBio Abcys) supplemented with 10% heat inactive FCS (EuroBio Abcys). The DLD1 BRCA2$^{-/-}$ cells were maintained in growth media containing 0.1 mg/ml hygromycin B (Thermo Fisher Scientific). The stable cell lines of DLD1$^{-/-}$ BRCA2 deficient cells expressing BRCA2 WT or variants of interest generated in this study were cultured in growth media containing 0.1 mg/ml hygromycin B and 1 mg/ml G418 (Sigma-Aldrich). All cells were cultured at 37 °C with 5% CO$_2$ in a humidified incubator and all cell lines used in this study have been regularly tested negatively for mycoplasma contamination.

For synchronization of cells in mitosis, nocodazole (100–300 ng/ml, Sigma-Aldrich) was added to the growth media and the cells were cultured for 14 h before harvesting. For synchronization by double thymidine block, the cells were treated with thymidine (2.5 mM, Sigma-Aldrich) for 17 h, released for 8 h followed by a second thymidine (2.5 mM) treatment for 15 h.

**Plasmids.** 2XMBP-, human 2XMBP-BRCA2$_{1–250}$ and EGFP-MBP-BRCA2 subcloning in phCMV1 expression vector were generated as described[41–48]. In the case of 2XMBP and 2XMBP-BRCA2$_{1–250}$, a tandem of 2 nuclear localization signals from RAD51 sequence was added downstream the MBP-tag.

Point mutations (M192T, S193A, S196N, S206C, and T207A) were introduced in the 2xMBP-BRCA2$_{1–250}$, EGFP-MBP-BRCA2 vector using QuikChange II and QuikChange XL site-directed mutagenesis kit (Agilent Technologies), respectively (see Supplementary Tables 3, 4 for primer sequences).

For expression of BRCA2$_{48–218}$ in bacteria, an optimized gene coding for human His-tagged BRCA2$_{48–218}$ (WT and T207A) was synthetized by Genscript and cloned in a pETM13 vector (a TEV site being present between the tag and the BRCA2 fragment). For expression of BRCA2$_{190–284}$ in bacteria, the human BRCA2$_{190–284}$ was amplified by PCR using full-length BRCA2 as template (phCMV1-2xMBP-BRCA2, see Supplementary Table 5 for primer sequences). The PCR product was purified and digested with BamH1 and SalI and cloned into the pGEX-6P-1 vector (GE Healthcare) to generate GST-BRCA2$_{190–284}$. The point mutation T207A was introduced in the same way in BRCA2$_{190–284}$ as in 2xMBP-BRCA2$_{1–250}$ and the EGFP-MBP-BRCA2. The introduction of the point mutation was verified by sequencing (see Supplementary Tables 3, 4 for primer sequences).

The PLK1 cDNA (Addgene pTK24) was cloned into the pFast-Bac HT vector using Gibson assembly (NEB) (see Supplementary Table 6 for primer sequences).

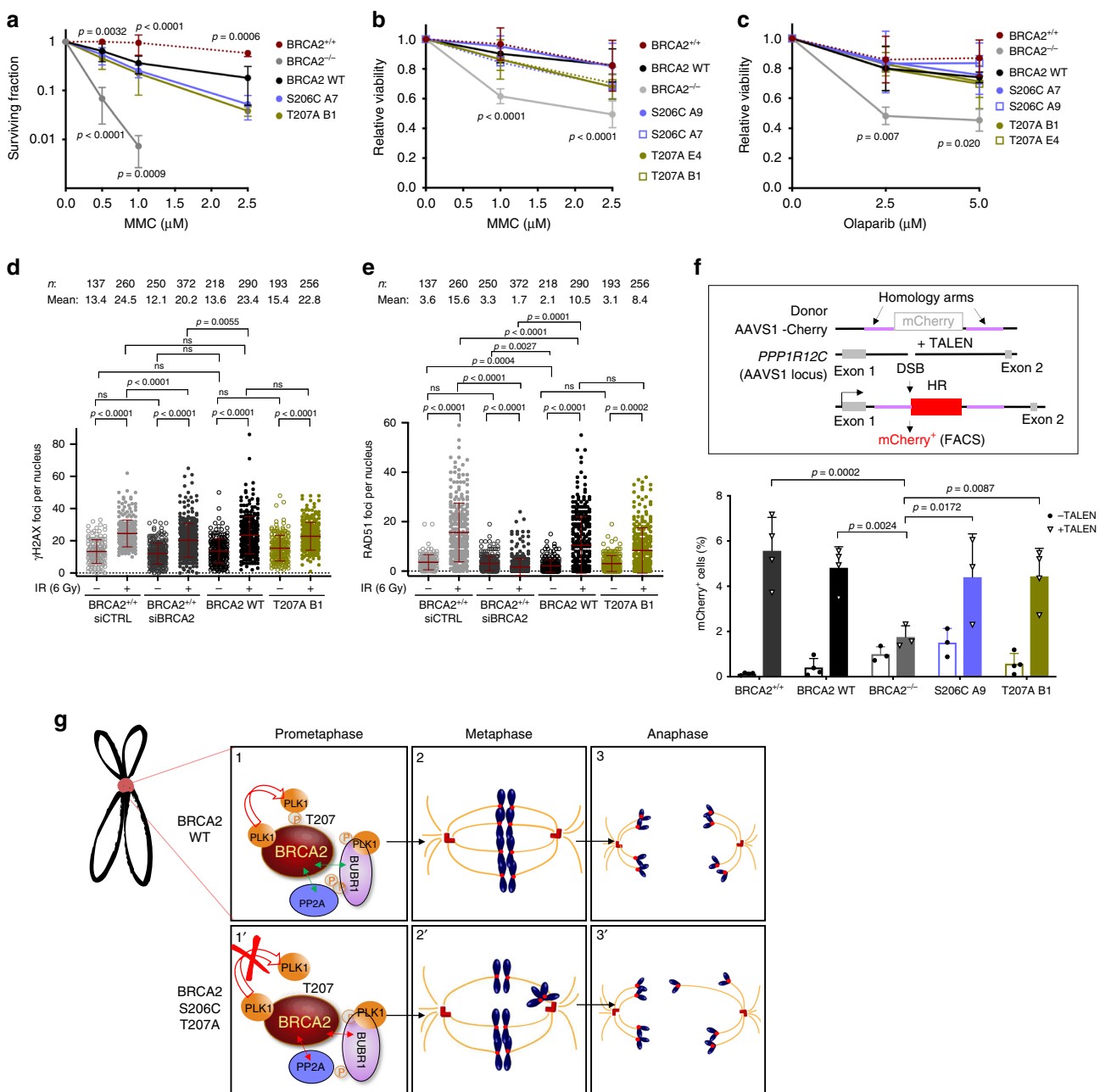

**Fig. 9 The DNA repair proficiency is not affected in cells bearing BRCA2 variants S206C and T207A. a** Quantification of the surviving fraction of BRCA2$^{+/+}$ or stable clones expressing BRCA2 WT or the variants S206C or T207A assessed by clonogenic survival upon exposure to MMC at concentrations: 0, 0.5, 1.0, and 2.5 μM. Data are represented as mean ± SD: BRCA2$^{+/+}$ (red) (n = 3), BRCA2$^{-/-}$ (gray) (n = 6), WT C1 (black) (n = 6), S206C A7 (blue) (n = 3), T207A B1 (green) (n = 4). **b–c** Quantification of the relative cell viability monitored by MTT assay upon treatment with increasing doses of MMC (**b**) or the PARP inhibitor Olaparib (**c**), as indicated. The data represent the mean ± SD of four (**b**) and three (**c**) independent experiments. **a–c** Two-way ANOVA test with Tukey's multiple comparisons test was used to calculate statistical significance of differences (the p-values show the significant differences compared to the BRCA2 WT clone). **d–e** Quantification of the number of γH2AX foci (**d**) or RAD51 foci per nucleus (**e**) 2 h after 6 Gy of γ-irradiation (+IR) versus non-irradiated conditions (−IR), in DLD1 BRCA2$^{+/+}$ cells depleted of BRCA2 (siBRCA2) or control cells (siCTRL) or cells bearing BRCA2 WT or the variant T207A. n indicates the total number of cells counted from two independent experiments. Kruskal-Wallis one-way analysis followed by Dunn's multiple comparison test was used to calculate statistical significance of differences. The red line in the plot indicates the mean ± SD, each dot represents a single focus. **f** Top: Scheme of the DSB-mediated gene targeting HR assay. Bottom: Frequency of mCherry positive cells in cells transfected with the promoter-less donor plasmid (AAVS1-2A-mCherry) without (−TALEN) or with (+TALEN) nucleases. The error bars represent mean ± SD of three to four independent experiments (BRCA2$^{+/+}$ (n = 4), WT (n = 4), BRCA2$^{-/-}$ (n = 3), S206C A9 (n = 3), T207A B1 (n = 4)). Two-way ANOVA test with Tukey's multiple comparisons test. **g** Model for the role of PLK1 phosphorylation of BRCA2 T207A by PLK1 in mitosis (see text for details). In panel 1 and 1′ the two-sided arrows represent a complex between the two proteins as indicated either favored by BRCA2 WT (green) or impaired in the BRCA2 mutated form (red); in panels 2, 2′, 3 and 3′ blue blobs represent chromosomes, red circles represent the kinetochores, red cylinders represent the centrioles and orange lanes represent the spindle microtubules. Source data are available as a Source Data file.

To produce PLK1-KD, the point mutation K82R was introduced in the pFast-Bac HT-PLK1 vector using QuikChange XL site-directed mutagenesis kit (see Supplementary Table 7 for primer sequences).

The Polo-like binding domain (PBD) of PLK1 (amino acid 326 to amino acid 603) was amplified from the pTK24 plasmid (Addgene) and cloned into a pT7-His6-SUMO expression vector using NEB Gibson assembly (Gibson Assembly Master Mix, New England BioLabs, Cat. #E2611S) (see Supplementary Table 7 for primer sequences). A plasmid containing a smaller PLK1 PBD fragment (amino acid 365 to amino acid 603) with a N-terminal GST tag was a kind gift from Dr. Anne Houdusse (Institute Curie, Paris).

To produce the phosphomimic BUBR1 mutant, we introduced the S670D, S676D and T680D point mutations in the pcDNA3-3xFLAG-BUBR1-RFP construct (kind gift from Dr. Geert JPL Kops) using QuikChange XL site-directed mutagenesis kit (see Supplementary Table 8 for primer sequences).

For the DSB-gene targeting assay, we replaced the GFP tag in the promoter-less AAVS1-2A-GFP-pA plasmid (kind gift from Dr. Carine Giovannangeli) with the mCherry tag from the pET28 mCherry plasmid using NEB Gibson Assembly (Gibson Assembly Master Mix, New England BioLabs, Cat. #E2611S). See Supplementary Table 11 for primer sequences.

**Expression and purification of 2xMBP-BRCA2$_{1-250}$.** The 2xMBP-BRCA2$_{1-250}$ was purified as previously described[49]. Briefly, ten 150 mm plates of HEK293T were transient transfected with the 2xMBP-BRCA2$_{1-250}$ using TurboFect (Thermo Fisher Scientific). The cells were harvested 30 h post-transfection, lysed in lysis buffer H (50 mM HEPES (pH 7.5), 250 mM NaCl, 1% NP-40, 5 mM EDTA, 1 mM DTT, 1 mM PMSF and EDTA-free Protease Inhibitor Cocktail (Roche)) and incubated with amylose resin (NEB) for 3 h at 4 °C. The 2xMBP-BRCA2$_{1-250}$ was eluted with 10 mM maltose. The eluate was further purified with Bio-Rex 70 cation-exchange resin (Bio-Rad) by NaCl step elution. The size and purity of the final fractions were analyzed by SDS-PAGE and western blotting using anti-MBP antibody. The 2xMBP-BRCA2$_{1-250}$ fragments containing the BRCA2 variants (M192T, S193A, S196N, S206C, and T207A) were purified following the same protocol as for WT 2xMBP-BRCA2$_{1-250}$.

**Expression and purification of BRCA2$_{48-218}$ and BRCA2$_{190-284}$ for NMR.** Recombinant $^{15}$N-labeled (WT, T207A) and $^{15}$N/$^{13}$C-labeled (WT) BRCA2$_{48-218}$were produced by transforming *Escherichia coli* BL21 (DE3) Star cells with the pETM13 vector containing human BRCA2$_{48-218}$ (WT and T207A). Recombinant $^{15}$N-labeled (WT, T207A) and $^{15}$N/$^{13}$C-labeled (WT, T207A) BRCA2$_{190-284}$ were produced by transforming *Escherichia coli* BL21 (DE3) Star cells with the pGEX-6P-1 vector containing human BRCA2$_{190-284}$ (WT and T207). Cells were grown in a M9 medium containing 0.5 g/l $^{15}$NH$_4$Cl and 2 g/l $^{13}$C-glucose when $^{13}$C labeling was needed. The bacterial culture was induced with 1 mM IPTG at an OD$_{600}$ of 0.8, and it was further incubated for 3 h at 37 °C. Harvested cells were resuspended in buffer A (50 mM Tris-HCl pH 8.0, 150 mM NaCl, 2 mM DTT, 1 mM EDTA) with 1 mM PMSF and 1X protease inhibitors cocktail (Roche) and disrupted by sonication. For BRCA2$_{48-218}$, clarified cell lysate was loaded onto Ni-NTA beads (Thermo Scientific) equilibrated with buffer A. After 1 h of incubation at 4 °C, beads were washed with buffer A containing 20 mM imidazole and eluted with buffer A containing 500 mM imidazole. The tag was cleaved by the TEV protease during a 2 hrs dialysis at 4 °C against 50 mM Tris-HCl pH 8.0, 150 mM NaCl, 1 mM EDTA 2 mM DTT. The sample was then boiled 10 min at 95 °C, spun down 5 min at 16,000 *xg* to remove thermosensitive contaminants and injected on Superdex 75 pg (GE Healthcare) equilibrated with 50 mM HEPES 1 mM EDTA pH 7.0. Sample concentration was calculated using its estimated molecular extinction coefficient of 9970 M$^{-1}$ cm$^{-1}$ at 280 nm. The protein sample was characterized for folding using NMR HSQC spectra, before and after the heating at 95 °C.

For BRCA2$_{190-284}$, clarified cell lysate was loaded onto Glutathione (GSH) Sepharose beads (GE Healthcare) equilibrated with buffer A. After 2 h of incubation at 4 °C, beads were washed with buffer A and eluted with buffer A containing 20 mM reduced glutathione. The tag was cleaved by the precision protease during an overnight dialysis at 4 °C against buffer B (50 mM HEPES pH 7.0, 1 mM EDTA) with 2 mM DTT and 150 mM NaCl. The cleaved GST-tag was removed by heating the sample for 15 min at 95 °C and spun it down for 10 min at 16,000g. Sample concentration was calculated using its estimated molecular extinction coefficient of 10,363 M$^{-1}$ cm$^{-1}$ at 280 nm. The protein sample was characterized for folding using NMR HSQC spectra, before and after the heating at 95 °C. BRCA2$_{190-284}$ was dialyzed overnight at 4 °C against buffer B with 2 mM DTT.

**Expression and purification of PLK1 and PLK1-kinase dead (PLK1-KD).** The recombinant 6xHis-PLK1 and 6xHis-PLK1-K82R mutant (PLK1-KD) were produced in sf9 insect cells by infection for 48 h (28 °C, 110 rpm shaking) with the recombinant baculovirus (PLK1-pFast-Bac HT vector). Infected cells were collected by centrifugation (1300 rpm, 10 min, 4 °C), washed with 1xPBS, resuspended in lysis buffer (1xPBS, 350 mM NaCl, 1% Triton X-100, 10% glycerol, EDTA-free Protease Inhibitor Cocktail (Roche), 30 mM imidazole). After 1 h rotation at 4 °C the lysate was centrifuged (25,000 rpm, 1 h, 4 °C) and the supernatant was collected, filtered (0.4 µm) and loaded immediately onto a Ni-NTA column (Macherey

Nagel) equilibrated with Buffer A1 (1xPBS with 350 mM NaCl, 10% glycerol and 30 mM imidazole, the column was washed with buffer A2 (1xPBS with 10% glycerol) and the protein was eluted with Buffer B1 (1x PBS with 10% glycerol and 250 mM imidazole). The eluted protein was diluted to 50 mM NaCl with Buffer A before being loaded onto a cationic exchange Capto S column (GE Healthcare) equilibrated with Buffer A1cex (50 mM HEPES (pH 7.4), 50 mM NaCl and 10% glycerol), the column was washed with Buffer A1cex before elution with Buffer B1cex (50 mM HEPES (pH 7.4), 2 M NaCl and 10% glycerol). The quality of the purified protein was analyzed by SDS-PAGE and the proteins concentration was determined using Bradford protocol with BSA as standard. The purest fractions were pooled and dialyzed against storage buffer (50 mM Tris-HCl (pH7.5), 150 mM NaCl, 0.25 mM DTT, 0.1 mM EDTA, 0.1 mM EGTA, 0.1 mM PMSF and 25% Glycerol) and stored in -80 °C. The purified proteins can be seen in Supplementary Fig. 10.

**Expression and purification of PLK1$_{PBD}$.** The pT7-6His-Sumo-PLK1 PBD (326-603) plasmid was expressed in Tuner pLacI pRare cells (Protein Expression and Purification Core Facility, Institut Curie), 2 L of TB medium with Kanamycin and Chloramphenicol antibiotics was inoculated with cells from the pre-culture. The cells were grown at 37 °C until an OD$_{600}$ of ~0.85. The temperature was decreased to 20 °C and the expression was induced by 1 mM IPTG overnight. The cells were harvested by 15 min of centrifugation at 4690g, at 4 °C. The cell pellets were suspended in 80 ml of 1 x PBS, pH 7.4, 150 mM NaCl, 10% glycerol, EDTA-free Protease Inhibitor Cocktail (Roche), 5 mM β-mercapto-ethanol (β-ME). The suspension was treated with benzonase nuclease and MgCl$_2$ at 1 mM final concentration for 20 min at 4 °C. The suspension was lysed by disintegration at 2 kbar (Cell distruptor T75, Cell D) followed by centrifugation at 43,000g, for 45 min, at 4 °C. The supernatant was loaded at 1 ml/min on a His-Trap FF-crude 5 mL column (GE Healthcare) equilibrated with PBS buffer, pH 7.4, 150 mM NaCl, 10% glycerol, 5 mM β-ME (A) and 20 mM imidazole. The proteins were eluted in a linear gradient from 0 to 100% with the same buffer (A) containing 200 mM imidazole, over 10 column volumes (CV). The purest fractions were pooled and dialyzed (8 kDa cut-off) against 20 mM Tris-HCl buffer, pH 8.0, 100 mM NaCl, 0.5 mM EDTA, 5 mM β-ME, 10% glycerol at 4 °C. 6xHis-SUMO Protease (Protein Expression and Purification Core Facility, Institut Curie) was added at 1/100 (w/w) and incubated overnight at 4 °C to cleave the 6xHis-SUMO tag. The cleaved PBD-PLK1 was purified using Ni-NTA agarose resin (Macherey Nagel), washed with the following buffer: 20 mM Tris-HCl pH 8.0, 100 mM NaCl, 0.5 mM EDTA, 5 mM β-ME and 10% glycerol. The sample was incubated with the resin for 1 h at 4 °C and the flow-through was collected. The sample was concentrated on an Amicon Ultra Centrifugal Filter Unit (10 kDa cut-off) and injected at 0.5 ml/min on a Hi-Load 16/60 Superdex column (GE healthcare), equilibrated with 20 mM Tris-HCl buffer, pH 8.0, 100 mM NaCl, 0.5 mM EDTA, 5 mM β-ME. The protein concentration was estimated by spectrophotometric measurement of absorbance at 280 nm. The purified protein is shown in Supplementary Fig. 10.

The GST-tagged PLK1$_{PBD}$ (365–603) was expressed in *E. coli* BL21 (DE3) STAR cells, induced with 0.5 mM IPTG at an OD$_{600}$ of 0.6, and grown at 37 °C for 3 h. The PBD (365-603) was purified by glutathione affinity chromatography. After GST cleavage (using a 6His-TEV protease), the tag and the protease were retained using GST- and NiNTA-agarose affinity chromatography, and the PBD collected in the flow-through was further purified by gel filtration chromatography. The protein was dialyzed against a buffer containing 50 mM Tris-HCl pH 8, NaCl 150 mM, and 5 mM β-ME.

**In vitro PLK1 kinase assay.** 0.5 µg purified 2xMBP-BRCA2$_{1-250}$ or 25 ng RAD51 protein, was incubated with recombinant active PLK1 (0, 50 or 100 ng) or PLK1-kinase dead (100 ng) (purchased from Abcam or purified from sf9 insect cells as detailed above, see Figure EV11B for the comparison of the kinase activity of both PLK1 preparations) in kinase buffer (25 mM HEPES, pH 7.6, 25 mM ß-glycer-ophosphate, 10 mM MgCl$_2$, 2 mM EDTA, 2 mM EGTA, 1 mM DTT, 1 mM Na$_3$VO$_4$, 10 µM ATP and 1 µCi [γ$^{32}$P] ATP (Perkin Elmer)) in a 25 µl total reaction volume. After 30 min incubation at 30 °C the reaction was stopped by heating at 95 °C for 5 min in SDS-PAGE sample loading buffer. The samples were resolved by 7.5% SDS-PAGE and [γ$^{32}$P] ATP labeled bands were analyzed with PhosphorImager (Amersham Bioscience) using the ImageQuant$^{TM}$ TL software (GE Healthcare Life Science). To control for the amount of substrate in the kinase reaction, before adding [γ$^{32}$P] ATP, half of the reaction was loaded on a 7.5% stain free SDS-PAGE gel (BioRad), the protein bands were visualized with ChemiDoc XRS + System (BioRad) and quantified by the Image Lab$^{TM}$ 5.2.1 Software (BioRad). The relative phosphorylation of 2xMBP-BRCA2$_{1-250}$ was quantified as $^{32}$P-labeled 2xMBP-BRCA2$_{1-250}$ (ImageQuant$^{TM}$ TL software) divided by the intensity of the 2xMBP-BRCA2$_{1-250}$ band in the SDS-PAGE gel (Image Lab$^{TM}$ 5.2.1 Software). In the control experiment where PLK1 inhibitor was used, 50 nM BI2536 (Selleck Chemicals) was added to the kinase buffer.

**In vitro protein binding assay.** To assess the interaction between recombinant PLK1 and BRCA2$_{1-250}$ after phosphorylation by PLK1, a kinase assay was performed with 0.2 µg recombinant PLK1 or PLK1-kinase dead (PLK1-KD) and 0.5 µg purified 2xMBP-BRCA2$_{1-250}$ (WT or the VUS T207) in kinase buffer supplemented

with 250 μM ATP (no [γ$^{32}$P] ATP) in a total reaction volume of 20 μl, one control reaction without ATP was performed with PLK1 and 2xMBP-BRCA2$_{1-250}$ WT. After 30 min incubation at 30 °C, 15 μl amylose beads was added to the reaction and incubated for 1 h at 4 °C. The beads were centrifuged at 2000$g$ for 2 min at 4 °C and the unbound fraction was collected before the beads were washed three time in kinase buffer (no ATP) containing 0.5% NP-40 and 0.1% Triton X-100. Bound proteins were eluted from the beads with 10 mM maltose, protein complexes were separated by SDS-PAGE and analyzed by western blotting. To control for the amount of proteins in the reaction, 2 μl of the kinase reaction (before adding the amylose beads) was loaded as input. The protein bands were visualized with ChemiDoc XRS + System (BioRad) and quantified by the Image Lab$^{TM}$ 5.2.1 Software (BioRad). The relative pull-down of PLK1 was quantified as the intensity of the PLK1 band in the pull-down divided by the intensity of the PLK1 band in the input (ImageQuant$^{TM}$ TL software).

To discard possible remaining phosphorylation of the 2xMBP-BRCA2$_{1-250}$ fragment coming from the purification of the protein from HEK293T cells, the 2xMBP-BRCA2$_{1-250}$ fragment (WT or the variant T207) was incubated with kinase buffer (no added ATP) supplemented with FastAP Thermosensitive Alkaline Phosphatase (Thermo Fisher Scientific Cat. #EF0654) for 1 h at 37 °C before addition of recombinant PLK1 followed by 30 min incubation at 30 °C and 1 h incubation with amylose beads as described above.

**NMR spectroscopy.** NMR experiments were carried out at 283 K on 600 and 700 MHz Bruker spectrometers equipped with a triple resonance cryoprobe. For NMR signal assignments, standard 3D triple resonance NMR experiments were recorded on $^{15}$N and $^{13}$C labeled samples of BRCA2$_{48-218}$ WT and BRCA2$_{190-284}$ WT and T207A. Analyses of these experiments provided backbone resonance assignment for the non-phosphorylated and phosphorylated forms of these BRCA2 fragments. To follow the PLK1 phosphorylation kinetics, the $^{15}$N labeled fragment BRCA2$_{48-218}$ (50 μM) was mixed to a first PLK1 sample at 0.1 μM and the $^{15}$N labeled fragment BRCA2$_{190-284}$ (200 μM) was mixed to another PLK1 sample at 1.1 μM. The mixes were incubated at pH 7.8 and 298 K. For each time point, a 140 μl sample was heated during 10 min at 368 K to inactivate PLK1, D$_2$O was added, the pH was adjusted to 7.0 and a $^{1}$H-$^{15}$N SOFAST-HSQC experiment $^{51}$ was recorded. The HSQC experiments were performed using 2048 ×256 time-points, 64 scans and an interscan delay of 80 ms. Data processing and analysis were carried out using the Topspin and CcpNmr Analysis 2.4.2 softwares.

**Analysis of phosphorylation assays followed by NMR.** In the HSQC spectra, the intensity of peaks of the phosphorylated residues pS193 and pT207, as well as the intensity of peaks corresponding to their non-phosphorylated form was retrieved at each time point of the kinetics. In order to estimate the fraction of phosphorylation for each residue at each point, the function Intensity$_{(phospho)}$ = f[Intensity$_{(non-phospho)}$] was drawn for each residue, the trendline was extrapolated to determine the intensity corresponding to the 100% phosphorylated residue and then the percentage of phosphorylation could be calculated at each time point by dividing peak intensities corresponding to the phosphorylated residue by the calculated intensity at 100% phosphorylation. Peaks corresponding to residues closed to a phosphorylated residue (L209 and V211 for pT207; D191, S197, and S195 for pS193) and thus affected by this phosphorylation were also treated using the same protocol and they were used to obtain a final averaged curve of the evolution of the percentage of phosphorylation at positions 193, 207 with time.

**Isothermal titration calorimetry.** ITC measurements were performed with the PLK1 PBD protein (amino acid 326 to amino acid 603) and BRCA2 peptides in 50 mM Tris-HCl buffer, pH 8.0 containing 150 mM NaCl and 5 mM β-ME, using a VP-ITC instrument (Malvern), at 293 K. We used automatic injections of 8 or 10 μl. The titration data were analyzed using the program Origin 7.0 (OriginLab) and fitted to a one-site binding model. To evaluate the heat of dilution, control experiments were done with peptide or protein solutions injected into the buffer. The peptides used for the ITC experiments were synthesized by GeneCust (Ellange, LU) or Genscript (Piscataway, NY). The peptides were acetylated and amidated at the N-terminal and C-terminal ends, respectively (see Supplementary Table 9 for peptide sequences). Only peptide BRCA2$_{190-284}$ was expressed in bacteria and purified as detailed above (see "Expression and purification of BRCA2$_{190-284}$ for NMR" section).

**Crystallization and structure determination.** The purified PBD protein (amino acid 365 to amino acid 603) was concentrated to 6 mg/ml, and mixed to the $^{194}$WSSSLATPPTLSS{pT}VLI$^{210}$ (pT207) BRCA2 peptide at a 3:1 molar ratio. The crystals were obtained by hanging drop vapor diffusion method at room temperature (293 K), by mixing 1 μl of complex with 1 μl of solution containing 10% PEG 3350, 100 mM BisTris pH 6.5, and 5 mM DTT. Diffraction data were collected at the Proxima 1 beamline (SOLEIL synchrotron, Gif-sur-Yvette, France). The dataset was indexed and integrated using XDS through the autoPROC package $^{52}$. The software performs an anisotropic cut-off (Tickle et al., STARANISO (2018) Global Phasing Ltd.) of merged intensity data, a Bayesian estimation of the structure amplitudes, and applies an anisotropic correction to the data. The structure was solved by molecular replacement using PHENIX (Phaser) software $^{53}$.

Two molecules of PBD were consecutively positioned. Electron density for the peptide was clearly visible in the position previously reported in other PBD structures in complex with phosphorylated peptides (PDB 4O56 or 3P35). Refinement was performed using BUSTER $^{54}$ and PHENIX $^{55}$. The model was built with Coot $^{56}$. A summary of crystallographic statistics is shown in Supplementary Table 2. The figures were prepared using Pymol v.1.7.4.0 (Schrödinger, LLC).

**Generation of stable DLD1 clones.** For generation of DLD1 BRCA2$^{-/-}$ cell lines stably expressing human BRCA2 variants of interest, we transfected one 100 mm plate of DLD1 BRCA2$^{-/-}$ cells at 70% of confluence with 10 μg of a plasmid containing human EGFP-MBP-tagged BRCA2 cDNA (corresponding to accession number NM_000059) using TurboFect (Thermo Fisher Scientific), 48 h post-transfection the cells were serial diluted and cultured in media containing 1 mg/ml G418 (Sigma-Aldrich) for selection. Single cells were isolated and expanded. To verify and select the clones, cells were resuspended in cold lysis buffer H (50 mM HEPES (pH 7.5), 250 mM NaCl, 1% NP-40, 5 mM EDTA, 1 mM DTT, 1 mM PMSF and EDTA-free Protease Inhibitor Cocktail (Roche), incubated on ice for 30 min, sonicated and centrifuged at 10,000$g$ for 15 min, 100 μg total protein lysate was run on a 4–15% SDS-PAGE followed by immunoblotting using BRCA2 and GFP antibodies to detect EGFP-MBP-BRCA2. Clones with similar expression levels were selected for functional studies.

The presence of the point mutations in the genome of the clones was confirmed by extraction of genomic DNA using Quick-DNA$^{TM}$ Universal Kit (ZYMO Research) followed by amplification of the N-terminal of BRCA2 (aa 1-267) by PCR using a forward primers that binds to the end of MBP and a reverse primer that binds to amino acid 267 in BRCA2, the presence of the point mutations was confirmed by sequencing of the PCR product (see Supplementary Table 4 and 10 for primer sequences).

**Cell extracts, immunoprecipitation and western blotting.** For the interaction between BRCA2$_{1-250}$ and endogenous PLK1, U2OS cells were transfected with 2xMBP-BRCA2$_{1-250}$ construct (WT, M192T, S193A, S196N, T200K, S206C, and T207A) using TurboFect (Thermo Fisher Scientific), 30 h post-transfection cells were synchronized by nocodazole (300 ng/ml), harvested and lysed in extraction buffer A (20 mM HEPES (pH 7.5), 150 mM NaCl, 0.1% NP40, 2 mM EGTA, 1.5 mM MgCl$_2$, 50 mM NaF, 10% glycerol, 1 mM Na$_3$VO$_4$, 20 mM ß-glycerophosphate, 1 mM DTT and EDTA-free Protease Inhibitor Cocktail (Roche). After centrifugation at 18,000$g$ for 15 min, the supernatant was incubated with amylose resin (NEB) for 1.5 h at 4 °C. The beads were washed five times in extraction buffer before elution with 10 mM maltose. Bound proteins were separated by SDS-PAGE and analyzed by western blotting. Where PLK1 and CDK1 inhibitor was used, the cells were synchronized in mitosis by nocodazole (14 h) followed by 2 h treatment with PLK1 inhibitor (50–100 nM BI2536 (Selleck Chemicals) or 50 μM BTO-1 (Sigma-Aldrich)) or the CDK1 inhibitor (10 μM, Ro-3306, (Selleck Chemicals)) before being harvested. The cells were lysed in extraction buffer A, pre-cleared by centrifugation and total protein lysate was separated by SDS-PAGE and analyzed by western blotting. Where proteasome inhibitor was used during the mitotic block, the cells were synchronized by nocodazole for 14 h before the MG-132 (50 μM, Sigma-Aldrich) was added to the media and the cells were cultured for additional 2 h before harvesting.

For analysis of pBUBR1, BUBR1, pT207-BRCA2 and BRCA2 levels in mitosis, nocodazole (100 ng/ml) treated DLD1 BRCA2$^{-/-}$ clones were lysed in extraction buffer A, pre-cleared by centrifugation and total protein lysate was separated by SDS-PAGE and analyzed by western blotting.

For analysis of the interaction between BRCA2-PLK1, BRCA2-BUBR1 and for the protein complex BRCA2-pBUBR1/BUBR1-PP2A(C)-PLK1 in mitosis, DLD1 BRCA2$^{-/-}$ stable clones expressing EGFP-MBP-BRCA2 (WT or the VUS S206C or T207A) were synchronized with nocodazole, harvested and lysed in extraction buffer A. The lysate were pre-cleared by centrifugation before incubation with GFP-TRAP beads (Chromotek) for 2 h at 4 °C to pull-down EGFP-MBP-BRCA2. Around 3 mg total protein lysate was used per pull-down. The beads were washed 5 times in extraction buffer A and 2 times in extraction buffer A with 500 mM NaCl. Bound proteins were eluted by boiling the samples for 4 min in 3x SDS-PAGE sample loading buffer (SB), eluted proteins were separated by SDS-PAGE and analyzed by western blotting using anti-mouse PLK1, anti-mouse BUBR1, anti-rabbit pT680-BUBR1, anti-mouse PP2A-C and anti-mouse BRCA2 (OP95) antibodies.

For immunoprecipitation of endogenous BUBR1, nocodazole treated DLD1 BRCA2$^{-/-}$ stable clones expressing BRCA2 WT or the variants (S206C or T207A) were lysed in extraction buffer A. After centrifugation, 2000–3000 μg total protein lysate was pre-cleared by incubation with 20 μl Protein G PLUS-Agarose (Santa Cruz, sc-2002) for 30 min at 4 °C. The pre-cleared lysate was incubated with 1.25 μg BUBR1 mouse antibody or control mouse IgG overnight at 4 °C before addition of 40 μl Protein G PLUS-Agarose, the lysate was incubated for additional 30 min before immunoprecipitates were collected by centrifugation. After four washes in extraction buffer A and two washes in extraction buffer A with 500 mM NaCl, the beads were re-suspended in SB, boiled and the immunocomplexes were analyzed by western blotting using anti-rabbit BUBR1, anti-mouse PLK1 and anti-mouse PP2A-C antibodies.

For the interaction between the phosphomimic BUBR1-3D mutant (S670D, S676D and T680D) and endogenous PP2A, the DLD1 BRCA2$^{-/-}$ stable clones expressing BRCA2 WT or the S206C variant was transient transfected with the pcDNA3-3xFLAG-BUBR1-3D-RFP construct. The transfection media was replaced 30 h post-transfection with fresh growth media containing 0.1 μg/ml nocadozole and the cells were incubated additional 14 h before harvesting. The cells were lysed and an immunoprecipitation was performed as described above for the BUBR1 immunoprecipitation using rabbit anti-tRFP antibody (Cat.#AB233, Evrogen) to pull-down the 3xFLAG-BUBR1-3D-RFP protein. Immunocomplexes were analyzed by western blotting using anti-mouse BUBR1 and anti-mouse PP2A-C antibodies.

For all Western blots, the protein bands were visualized with ChemiDoc XRS + System (BioRad) and quantified by the Image Lab$^{TM}$ 5.2.1 Software (BioRad). For the relative expression levels (Fig. 5g-i, Supplementary Fig. 5d), the intensity of the band of interest was divided by the intensity of the signal from the stain free gel. The results are presented as percentage compared to BRCA2 WT clone. To calculate the relative co-immunoprecipitation (co-IP)/co-pull-down of a protein of interest, the intensity of the band in the co-IP was divided by the intensity of the band in the input (ImageQuant$^{TM}$ TL software), the ratio co-IP:input of the protein of interest was then divided by the intensity of the band of the immunoprecipitated protein. StainFree images of the gels before transfer were used as loading control for the input and cropped image is shown in the figures.

**Antibodies used for western blotting**. mouse anti-MBP (1:5000, R29, Cat. #MA5-14122, Thermo Fisher Scientific), mouse anti-BRCA2 (1:1000, OP95, EMD Millipore), rabbit anti-pT207-BRCA2 (raised for this study using the peptide $^{203}$TLSS-pT-VLIVRNEEAC as antigen, Genscript) (1:1000), anti-GFP (1:5000, Protein Expression and Purification Core Facility, Institut Curie), mouse anti-PLK1 (1:5000, clone 35-206, Cat. #05-844, EMD Millipore), mouse anti-BUBR1 (1:1000, Cat. #612502, BD Transduction Laboratories), rabbit anti-BUBR1 (1:2000, Cat. #A300-386A, Bethyl Laboratories), mouse anti-PP2A C subunit (1:1000, clone 1D6, Cat. #05-421, EMD Millipore), rabbit anti-pT680-BUBR1 (1:1000, EPR 19958, Cat. #ab200061, Abcam), and rabbit anti-pS676-BUBR1 (1:1000, R193, kind gift from Dr. Erich A. Nigg). Horseradish peroxidase (HRP) conjugated 2nd antibodies used: mouse-IgGκ BP-HRP (IB: 1:10 000, Cat. #sc-516102, Santa Cruz), goat anti-rabbit IgG-HRP (IB: 1:5000, Cat. #sc-2054, Santa Cruz), goat anti-mouse IgG-HRP (1:10 000, Cat.#115-035-003, Interchim), goat anti-rabbit IgG-HRP (1:10 000, Interchim, Cat.#111-035-003).

**siRNA transfection**. For analysis of the pT680-BUBR1 levels in U2OS cells after transient depletion of endogenous BRCA2 with RNAi, U2OS cells were transfected with 200 nM siRNA targeting the 3′UTR of BRCA2 (siBRCA2 #1: SI00000966, Qiagen) using jetPRIME (Polyplus Transfection, Cat.#114-07). As control, the cells were transfected with the 200 nM of the si-control RNA (siRNA control ON-TARGETplus Non-targeting Pool. D-001810-10-05, Thermo Scientific) The transfection media was replaced 30 h post-transfection with fresh growth media containing 0.1 μg/ml nocadozole and the cells were incubated additional 14 h before harvesting. Total protein lysate was extracted as described above and 30–50 μg total protein lysate was resolved on a 4–15% SDS-PAGE and analyzed by western blotting using anti-mouse BRCA2 (OP95), anti-mouse BUBR1, anti-rabbit pT680-BUBR1 and anti-mouse PLK1 antibodies (see above for reference number).

For depletion of endogenous BRCA2 in DLD1 BRCA2$^{+/+}$ cells for analysis of γH2AX and RAD51 foci, the cells were transfected with a combination of the BRCA2 5′UTR siRNA (SI00000966, Qiagen) and IAC204 (Dharmacon D-003462-04) (100 nM each) or the ON-TARGET plus Non-targeting oligonucleotide D-001810-04-20, Thermo Scientific 100 nM) and fixed and expose to IR 30 h post-transfection (see Supplementary Fig. 8c for the Western blot showing BRCA2 depletion) for analysis of DNA repair foci (see Supplementary Fig. 8c for the Western blot showing BRCA2 depletion).

**Phosphatase treatment**. DLD1 BRCA2$^{-/-}$ cells stably expressing EGFP-MBP-BRCA2 WT were synchronized in mitosis by nocodazole (14 h), harvested, lysed in extraction buffer A without phosphatase inhibitors (NaF, Na$_3$VO$_4$ and ß-glycer-ophosphate), and pre-cleared by centrifugation. For detection of pT207-BRCA2 and BRCA2, 20 U FastAP Thermosensitive Alkaline Phosphatase (Thermo Fisher Scientific Cat. #EF0654) was added to 200 μg of total protein lysate in FastAP Buffer in a total reaction volume of 60 μl. After 1 h incubation at 37 °C the reaction was stopped by heating at 95 °C for 5 min in SDS-PAGE sample loading buffer, 30 μl of the reaction was loaded on a 4–15% SDS-PAGE gel, the gel was transferred onto nitrocellulose membrane and the levels of pT207-BRCA2 were analyzed by western blotting using anti-pT207-BRCA2 antibody. For detection of pBUBR1 and BUBR1, increased amount (0-20U) of FastAP Thermosensitive Alkaline Phosphatase was added to 15 μg of total protein lysate in FastAP Buffer in a total reaction volume of 60 μl followed by same protocol as described for pT207-BRCA2, the levels of pS676/pT680-BUBR1 were analyzed by western blotting using anti-pS676/pT680-BUBR1 antibodies.

**Cell survival and viability assays**. For clonogenic survival assay, DLD1 BRCA2$^{-/-}$ cells stably expressing full-length EGFP-MBP-BRCA2 and the variants (S206C

and T207A) were treated at 70% of confluence with Mitomycin C (Sigma-Aldrich) at concentrations: 0, 0.5, 1.0, and 2.5 μM. After 1 h drug treatment the cells were serial diluted in normal growth media containing penicillin/streptomycin (Euro-bio) and seeded in triplicates into 6-well plates. The media was changed every third day, after 10–12 days in culture the plates were stained with crystal violet, colonies were counted and the surviving fraction was determined for each drug concentration.

Cell viability was assessed with 3-[4,5-Dimethylthiazol-2-yl]-2,5-diphenyltetrazolium bromide (MTT, #M5655, Sigma Aldrich) after treatment with MMC and the PARP inhibitor Olaparib (AZD2281, Ku-0059436, #S1060, Selleck Chemicals). For MMC, the cells were plated in triplicates in 96-well microplates (3000–5000 cells/well) the day before treatment. The cells were washed once in PBS before addition of serum-free media containing MMC at the concentrations: 0, 1.0 and 2.5 μM. After 1 h treatment the cells were washed once in PBS and incubated for 72 h in normal growth media before the viability was measured by MTT assay. For PARP inhibition, the cells were seeded 4 h before 4 days treatment in normal growth media with Olaparib at concentrations: 0, 2.5 and 5.0 μM.

**HR assays**. We applied a DSB-mediated gene targeting strategy using site-specific TALEN nucleases to quantify HR in cells. DLD1 BRCA2$^{-/-}$ cells stably expressing full-length GFPMBP-BRCA2 and the variants (S206C and T207A) were transfected using AMAXA technology (Lonza) nucleofector kit V (Cat. #VCA-1003) with 3 μg of the promoter-less donor plasmid (AAVS1-2A-mCherry) with or without 1 μg of each AAVS1-TALEN encoding plasmids (TALEN-AAVS1-5′ and TALEN-AAVS1-3′, see Supplementary Table 12 for sequences, kind gift from Dr. Carine Giovannangeli). For each transfection, $1 \times 10^6$ cells were transfected using program L-024, the cells were seeded in 6-well plate in culture media without selection antibiotics. The day after transfection the media was changed to media with selection and 48 h post-transfection the cells were trypsinized and reseeded on a 10-cm culture dish and cultured for additional 5 days. The percentage of mCherry positive cells was analyzed on a BD FACSAria III (BD Bioscience) using the FACSDiva software and data were analyzed with the FlowJo 10.4.2 software (Tree Star Inc.).

**Analysis of tetraploid cells**. For the analysis of S-phase tetraploid cells in the DLD1 BRCA2$^{-/-}$ stable clones, the cells were incubated with 10 μM BrdU for 20 min before they were harvest, fixed and stained for cell cycle analysis using a APC-BrdU flow kit (BD Bioscience, Cat. #552598) following the manufacturer's instructions.

Labeled cells were analyzed on a BD FACSCanto II (BD Bioscience) using the FACSDiva software and data were analyzed with the FlowJo 10.4.2 software (Tree Star Inc.).

**Immunofluorescence**. Kinetochore localization: For staining of pT680-BUBR1 and PLK1 at the kinetochore, DLD1 BRCA2$^{-/-}$ stable clones expressing EGFP-MBP-BRCA2-WT or the variant T207A were seeded on coverslips and treated with nocodazole (0.25 μg/ml) for 4 h, fixed with 4% PFA in PBS containing 0.5% Triton X-100 for 20 min at room temperature. The coverslips were rinsed three times in PBS-T and blocked for 30 min with 4% BSA in PBS before incubation with primary antibodies (human anti-CREST (1:100, Cat. #15-234-0001, Antibodies Online) together with either rabbit anti-pT680-BUBR1 (1:500, clone EPR 19958, Abcam, Cat. #ab200061) or mouse anti-PLK1 (1:500, clone F-8, Santa Cruz Biotechnology, Cat. #sc-17783), diluted in PBS-T with 5% BSA overnight at 4 °C. After three washes of 5 min in PBS-T the coverslips were incubated for 2 h incubation at room temperature with respective Alexa Fluor conjugated secondary antibody (for pBUBR1; goat anti-human Alexa-488 (1:1000, Cat. #A11013, Life Technologies) and donkey anti-rabbit Alexa-488 (1:1000, Cat. #A-21206, Thermo Fisher Scientific), for PLK1; goat anti-human Alexa-633 (1:500, Cat. #A21091, Life Technologies) together with either donkey anti-rabbit Alexa-488 (1:1000, Cat. #A-21206, Thermo Fisher Scientific) or donkey anti-mouse Alexa-488 (1:1000, Cat. #A-21202, Thermo Fisher Scientific) diluted in PBS-T with 5% BSA. After two washes of 5 min in PBS-T and one rinse in PBS the coverslips were mounted on microscope slides.

For staining of BRCA2 at the kinetochore, U2OS was transient transfected with GFPMBP-BRCA2 construct using TurboFect (Thermo Fisher Scientific). The cells were seeded on coverslips 24 h post-transfection and incubated for another 24 h before nocodazole (0.25 μg/ml) was added. The cells were treated for 4 h followed by fixation as described above for kinetochore localization of pBUBR1, pAuroraB and PLK1. BRCA2 was detected by rabbit anti-BRCA2 (1:500, CA1033, EMD Millipore) and Alexa-488 secondary antibody (donkey anti-rabbit Alexa-488 (1:1000, Cat. #A-21206, Thermo Fisher Scientific), CREST was detected by human anti-CREST (1:100, Cat. #15-234-0001, Antibodies Online) and Alexa-633 secondary antibody (1:500, Cat. #A21091, Life Technologies)), diluted in PBS-T with 5% BSA.

Phosphatase inhibitors (50 mM NaF, 1 mM Na$_3$VO$_4$, and 20 mM ß-glycerophosphate) were added to all buffers.

Cold stable kinetochore–microtubules, chromosome alignment and segregation: DLD1 BRCA2$^{-/-}$ cells and the stable clones expressing EGFP-MBP-BRCA2 WT or the variants (S206C and T207A) were seeded on coverslips in 6-well tissue

culture plates and synchronized in mitosis. For analysis of chromosome alignment, the cells were synchronized by double thymidine (2.5 mM, Sigma-Aldrich) block, released for 9 h followed by treatment with Monastrol (100 μM, Sigma-Aldrich) for 16 h. After incubation with Monastrol the cells were washed twice in PBS before 1 h incubation in media containing the proteasome inhibitor MG-132 (10 μM, Sigma-Aldrich). To detect cold-stable microtubules cells were synchronized using the same protocol as for the chromosome alignment followed by 15 min of cold treatment on ice before fixation. For chromosome segregation analysis, the cells were synchronized by double thymidine block and released in normal growth media for 11 h.

After synchronization, the cells were fixed with 100% methanol for 15 min at -20 °C, rinsed once in PBS before permeabilization with PBS containing 0.1% Triton-X for 15 min at room temperature. Nonspecific epitope binding was blocked with 4% BSA (Sigma-Aldrich) in PBS. The coverslips were rinsed in PBS, incubated with primary antibody (mouse anti-α-tubulin (1:5000, GT114, Cat. #GTX628802, Euromedex) and human anti-CREST (1:100, Cat. #15-234-0001, Antibodies Online)) diluted in PBS containing 0.1% Tween-20 (PBS-T) and 5% BSA for 1 h at room temperature. After incubation, the coverslips were washed three times of 5 min in PBS-T before being incubated for 1 h at room temperature with Alexa Fluor conjugated secondary antibody (donkey anti-mouse Alexa-594 (1:1000, Cat. #A-21203, Thermo Fisher Scientific) and goat anti-human Alexa-488 (1:1000, Cat. #A11013, Life Technologies)) diluted in PBS-T with 5% BSA. The coverslips were washed two times of 5 min each in PBS-T followed by one rinse in PBS before being mounted on microscope slides.

For the analysis of chromosome alignment in cells expressing the phosphomimic BUBR1-3D mutant (S670D, S676D and T680D), the DLD1 BRCA2$^{-/-}$ stable clones expressing the T207A variant was transient transfected with the pcDNA3-3xFLAG-BUBR1-3D-RFP construct using TurboFect. The day after transfection the cells were seeded on coverslips in 6-well tissue culture plates, synchronized in mitosis and prepared for immunofluorescence.

Aneuploidy: For aneuploidy analysis the cells were treated with nocodazole for 14 h (0.1 μg/ml) to enable chromosome spread. The cells were rinsed in PBS, incubated for 10 min with KCl (50 mM) at room temperature before they were spread on coverslips at 900 rpm for 5 min in a Cytospin 4 (Thermo Scientific). The cells were fixed with 3% paraformaldehyde (PFA) in PBS for 20 min followed by 15 min permeabilization in PBS containing 0.1% Triton X-100. The coverslips were rinsed three times in PBS, blocked with 5% BSA in PBS before incubation with human anti-CREST primary antibody (1:100, Cat. #15-234-0001, Antibodies Online) diluted in PBS overnight at 4 °C. After incubation the coverslips were washed three times of 5 min in PBS before 1 h incubation at room temperature with Alexa Fluor conjugated secondary antibody) (goat anti-human Alexa-555 (1:1000, Cat. #A-21433, Thermo Fisher Scientific)) diluted in PBS. After three washes of 5 min in PBS the coverslips were mounted on microscope slides.

γH2AX and RAD51 foci: For the detection of γH2AX and RAD51 foci, the cells were seeded on coverslips the day before 6 Gy γ-irradiation (GSR D1, Cs-137 irradiator). Two hours after irradiation, the coverslips were washed twice in PBS followed by one wash in CSK Buffer (10 mM PIPES, pH 6.8, 0.1 M NaCl, 0.3 M sucrose, 3 mM MgCl₂, EDTA-free Protease Inhibitor Cocktail (Roche)). The cells were permeabilized for 5 min at room temperature in CSK buffer containing 0.5% Triton X-100 (CSK-T) followed by one rinse in CSK buffer and one rinse in PBS before fixation for 20 min at room temperature with 2% PFA in PBS. After one rinse in PBS and one in PBS-T, the cells were blocked for 5 min at room temperature with 5% BSA in PBS-T before incubation for 2 h at room temperature with primary antibodies diluted in PBS-T with 5% BSA. After primary antibody incubation, the coverslips were rinsed in PBS-T followed by two washes of 10 min in PBS-T and blocked for 5 min at room temperature with 5% BSA in PBS-T before incubation for 1 h at room temperature with respective Alexa Fluor conjugated secondary antibody diluted in PBS-T with 5% BSA. After one rinse in PBS-T and two washes of 10 min in PBS-T the coverslips were rinsed in PBS before being mounted on microscope slides. γH2AX foci were detected by mouse anti-pSer139-γH2AX (1:1000, clone JBW301, EMD-Millipore, Cat. #05-636) and secondary antibody donkey anti-mouse Alexa-594 (1:1000, Cat. #A-21203, Thermo Fisher Scientific). RAD51 foci were detected by rabbit anti-RAD51 (1:100, clone H-92, Santa Cruz Biotechnology, Cat. #sc-8349), followed by secondary antibody donkey anti-rabbit Alexa-488 (1:1000, Cat. #A-21206, Thermo Fisher Scientific).

For the analysis of γH2AX and RAD51 foci in DLD1 BRCA2$^{+/+}$ cells depleted of BRCA2 by siRNA, the DLD1 BRCA2$^{+/+}$ cells were transient transfected with siRNA targeting BRCA2 (see section siRNA transfection above). The day after transfection the cells were seeded on coverslips in 6-well tissue culture plates and radiated as described above.

Micronuclei: For analysis of micronuclei, DLD1 BRCA2$^{-/-}$ cells and the stable clones expressing EGFP-MBP-BRCA2 WT or the variants (S206C and T207A) were seeded on coverslips in 6-well tissue culture plates the day before fixation. Centromeres were detected by human anti-CREST primary antibody (1:100, Cat. #15-234-0001, Antibodies Online) and Alexa Fluor conjugated secondary antibody (goat anti-human Alexa-555 (1:1000, Cat. #A-21433, Thermo Fisher Scientific)).

All coverslips were mounted on microscope slides with ProLong Diamond Antifade Mountant with DAPI (Cat. #P36966, Thermo Fisher Scientific).

Image acquisition and analysis: For analysis of DNA repair foci, chromosome alignment and segregation, images were acquired in an upright Leica DM6000B wide-field microscope equipped with a Leica Plan Apo 63x NA 1.4 oil immersion objective. The camera used is a Hamamatsu Flash 4.0 sCMOS controlled with the MetaMorph2.1 software (Molecular Devices). For Fig. 6a and 7a, 7 to 20 Z-stacks were taken at 0.2 μm intervals to generate a maximal intensity projection image using ImageJ. For the analysis of γH2AX and RAD51 foci, 26 Z-stacks were taken at 0.2 μm intervals to generate a maximal intensity projection using the Image J software (1.51 s, NIH). For the BUBR1-3D-RFP chromosome alignment experiment, RFP negative cells on the same coverslips were used as control.

The number of γH2AX foci per nucleus were counted by a customized macro using a semi-automated procedure; the nucleus was defined by an auto-threshold (Otsu, Image J) on DAPI, a mask was generated and applied onto the Z-projection to count foci within the nucleus. For the definition of foci we applied the threshold plugin IsoData (ImageJ software (1.51 s, NIH)) and for the quantification of foci we used the tool Analyze Particles (ImageJ software (1.51 s, NIH)) setting a range of 5-100 pixels² to select only particles that correspond to the size of a focus. RAD51 foci were quantified using the plugin Find Maxima onto the Z-projection with a prominence of 1000.

For analysis of cold stable microtubules images were acquired in an upright Leica DM6000B wide-field microscope equipped with a Leica Plan Apo 100x NA 1.4 oil immersion objective. The camera used is a Hamamatsu Flash 4.0 sCMOS controlled with the MetaMorph2.1 software (Molecular Devices), 20 Z-stacks were taken on metaphase cells at 0.5 μm intervals to generate a maximal intensity projection image using the ImageJ software (1.51 s, NIH). The quantification of the intensity of α-tubulin in metaphase cells was performed in the area of the spindle subtracting the background of α-tubulin signal from a different area in the same cell. The data were then normalized to the mean intensity of the BRCA2 WT cells.

For analysis of aneuploidy, kinetochore localization and micronuclei images were acquired in an inverted confocal Leica SP5 microscope with a plan Apo 63x NA 1.4 oil immersion objective with the lasers 405, 488, 561 and 633 nm. For Fig. 8b (aneuploidy), Z-stacks were taken at 0.13 μm intervals to generate a maximal intensity projection image using the ImageJ software (1.51 s, NIH). For the counting of chromosomes in the aneuploidy experiment, the quantification was performed in zoomed areas counting the CREST signal in separated stacks to ensure the counting of all chromosomes. We were able to count up to 65 chromosomes with certainty, thus >65 CREST signals were discarded and not included in the analysis.

For the analysis of kinetochore localization, Z-stacks were taken at 0.21 μm intervals to generate a sum slice projection image using ImageJ, six pairs of chromosomes per cell were analyzed in 15-21 cells per experiment from two individual experiment. The results from the quantifications (Fig. 5i, Supplementary Fig. 6b) is represented as the ratio between the intensities for pBUBR1/PLK1 and the CREST signal relative to the mean ratio observed for the BRCA2 WT complemented cells. For the images in Fig. 5h (pBUBR1:CREST), Supplementary Fig. 6a (PLK1:CREST) and Supplementary Fig. 8e (micronuclei), Z-stacks were taken to generate a maximal intensity projection image using the ImageJ software (1.51 s, NIH), except for DAPI where the image is from one Z-stack.

**Time-lapse video microscopy of mitotic cells.** For phase-contrast video-microscopy DLD1 BRCA2$^{-/-}$ cells stably expressing full-length EGFP-MBP-BRCA2 and the variants (S206C and T207A) were seeded in 35 mm Ibidi μ-Dishes (Ibidi, Cat. #81156), synchronized by double thymidine block, released and cultured for 4 h in normal growth media before the filming was started. The cells were imaged for 16 h every 5 min, at oil-40X using an inverted video-microscope (Leica DMI6000) equipped with electron multiplying charge coupled device (EMCCD) camera controlled by the MetaMorph2.1 software (Molecular Devices). Images were mounted using the Image J software (1.51 s, NIH).

**Statistical analysis.** In all graphs error bars represent the standard deviation (SD) from at least three independent experiments unless otherwise stated, scatter dot plots show median with 95% CI. Statistical significance of differences was calculated with unpaired two-tailed $t$-test, one/two-way ANOVA with Dunnett's or Tukey's multiple comparisons test, Mann–Whitney two-tailed or Kruskal–Wallis one-way analysis followed by Dunn's multiple comparisons test as indicated in the figure legends. All analyses were conducted using GraphPad Prism version Mac OS X 8.3.0 (328).

**Reporting summary.** Further information on research design is available in the Nature Research Reporting Summary linked to this article.

## Data availability

The data that support the findings of this study are available from the corresponding authors upon reasonable request. The source data for Fig. 1a–d; Fig. 2; Fig. 3a–k; Fig. 4a–f; Fig. 5a–i; Figs. 6a, 6d; Figs. 7a, c, d; Fig. 8a, d; Fig. 9a–f; Supplementary Fig. 1a, b; Supplementary Figu. 2b, c; Supplementary Fig. 3a–d; Supplementary Fig. 4a, c; Supplementary Fig. 5b–d; Supplementary Fig. 6b–d; Supplementary Fig. 7d; Supplementary Fig. 8c–d; Supplementary Fig. 10a, b; are available as a Source Data file.

The PDB files used during structure analysis can be retrieved using the PDB codes 4O56 and 3P35 on the PDB website: https://www.rcsb.org. The final X-ray structure of the PLK1 PBD bound to the phosphorylated BRCA2 peptide is also available on this website, under the code 6GY2.

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

## Acknowledgements

The authors thank members of the AC lab for fruitful comments on the manuscript and Davide Panigada and Jose M. Jimenez-Gomez for the illustration of chromosomes in Fig. 9. We thank Rene H. Medema for useful discussions on this work including the cell synchronization protocol used in Fig. 7. We also thank Juan S. Martinez for construct BRCA2$_{1-250}$, Anne Houdusse for construct PLK1365-603, Carine Giovannangeli for TALEN plasmids, Eric Nigg for pS676-BUBR1 antibody and Geert JPL Kops for BUBR1-RFP construct. We acknowledge the Cell and Tissue Imaging Facility of the Institut Curie (PICT), a member of the France BioImaging National Infrastructure (ANR-10-INBS-04), and the French Infrastructure for Integrated Structural Biology (https://www.structuralbiology.eu/networks/frisbi, ANR-10-INSB-05-01). We thank Charlene Lasgi from the Flow Cytometry platform of Institut Curie, Orsay. We thank Guillaume Hoffmann and Jose A. Marquez from the HTXLab (Grenoble, France), funded by the European Community's Seventh Framework Programme H2020 under iNEXT (grant agreement No. 653706). This work was supported by the ATIP-AVENIR CNRS/INSERM Young Investigator grant 201201, FRM "Amorcage Jeunes Equipes" Young Investigator grant AJE20110 and Institut National du Cancer INCa-DGOS_8706 to A.C. and S.Z.J.; A. E. was supported by the Swedish Society for Medical Research.

## Author contributions

A.E. purified WT and mutated BRCA2$_{1-250}$, established the stable DLD1$^{-/-}$ cell lines, performed kinase assays, pull-down assays, Western blots, time-lapse microscopy experiments, mitotic index measurements by FACS, clonogenic survival and MTT assays as well as the statistical analysis for all the experiments. C.M. performed Western blot of pT207, IF and image acquisition of cold-stable microtubules, metaphase plate alignment, chromosome segregation and karyotype analysis. M.J. performed the NMR experiments

assisted by S. M., F.T., and S.Z.J. M.J. and S.M. purified PLK1$_{PBD}$ and performed the ITC experiments. S.M. solved the X-ray structure assisted by V.R. R.B. and G.S. performed IF, image acquisition and quantification of DNA repair foci. V.B. assisted establishing stable clones and performing clonogenic survival assays. P.D. purified PLK1$_{PBD}$. A.M. cloned and produced PLK1 and PLK1-KD from insect cells. A.C., A.E., and S.Z.J. designed the experiments. A.C. and S.Z.J. supervised the work. A.C. wrote the paper with important contributions from all authors.

## Competing interests
The authors declare no competing interests.
