## [Peer Review File · Nature Communications]

Reviewers' comments:

Reviewer #1 (Remarks to the Author):

The manuscript is novel, very exciting, well-written, technically sound, a piece of work of high quality and should be considered for publication. The authors identified a direct mitotic role of BRCA2 in the alignment of chromosomes separate from its well-studied role in DNA repair function with direct consequences on chromosome stability. Thus, it is very interesting for a broad readership. For me as a non-expert in mitosis the manuscript was very clear and its biological implications well understandable. The length of the text is appropriate.

Reviewer #2 (Remarks to the Author):

In this manuscript that nicely combines structural and function studies, Ehlén et al. provide strong evidence that phosphorylation of DNA damage response protein BRCA2 by Polo-like kinase PLK1 plays a critical and unexpected role in regulating mitosis. Using multiple approaches, it is shown that two sites in BRCA2, S193 and T207, are phosphorylated and that recognition of pT207 by PLK1 Polo-box domain primes the interaction of PLK1 with BRCA2. Of note, the T207A and S206C variants have been found in breast cancer patients and were shown by the authors to cause defective chromosome congression during mitosis and to delay mitosis progression. The NMR approach used to unambiguously identify the phosphorylated BRCA2 sites in vitro is particularly exciting. The interaction of PLK1 with pT207 BRCA2 was quantified by calorimetry and X-ray crystallography. Using biochemistry and cell biology approaches, it is shown that the PLK1/BRCA2 complex associates with PP2A and phosphorylated BUBR1 and that the ensuing tetrameric protein assembly contributes to proper chromosome alignment during mitosis. Importantly, data presented in the manuscript suggest that the PLK1/BRCA2 interaction does not play a role in the well-established homologous recombination DNA repair function of BRCA2. While this reviewer is less qualified to comment on the cell biology aspects of the study, the work is clearly of high quality. The conclusions are well supported by the data presented in the manuscript. The manuscript is well organized and well written.

Reviewer #3 (Remarks to the Author):

Proper chromosome alignment depends on BRCA2 phosphorylation by PLK1

Ehlen et al.

The authors address in this manuscript whether phosphorylation of the tumour suppressor BRCA2 by PLK1 is important for mitosis. The authors start by asking whether missense variants in BRCA2 from cancer patients are phosphorylated differently by PLK1 compared to WT BRCA2. Indeed, they find that T207A and S206C in BRCA2 decrease PLK1 phosphorylation in vitro two-fold. Next, the authors use NMR in order to quantify PLK1 phospho-sites in BRCA2 fragments. This analysis shows that the kinetics of S193 phosphorylation is dependent on T207. The authors test the model that T207 is important for the shelf-priming of PLK1 binding. In agreement with this model, T207A and S206C mutant fragments were less efficient in pulling down PLK1 than the phosphorylated WT peptide. Next they analyse the binding of the BRCA2 peptide to the PLK1PBD by crystallography providing a structural understanding of this interaction.

The authors then study the consequences of BRCA2 phosphorylation by PLK1 in vivo using a BRCA2 deficient human cell line. They provide evidence of an additional PLK1 binding site that is primed by CDK1 phosphorylation of T77. Interestingly, the duration of mitosis for S206C and T207A cells is longer than for WT. This mitotic delay was due to a reduction in the formation of a complex between BRCA2, PLK1, BUBR1 and PP2A in the S206C and T207A mutants. In addition, pBUBR1 at kinetochores was reduced in T207A cells. This defect accumulated in chromosome

missegregation errors, mostly in lagging chromosomes and in high numbers of aneuploidy. Finally, the authors convincingly show that the delay in mitosis is not an indirect consequence of DNA damage and that DNA repair was not grossly affected by BRCA2-T207A.

Taken together, this is an interesting manuscript of overall high quality that describes an important function of BRCA2 in complex with PP2A, BUBR1 and PLK1 in mitosis. Disturbing this function of BRCA2, probably in regulating kinetochore-microtubule interactions, leads to chromosome missegregation without having a strong impact on the function of BRCA2 in DNA repair. I support publication of this manuscript in Nat. Comm. after revision.

Specific points

1. What are the evidences that T297 is phosphorylated by PLK1 in vivo? Was this P-site identified in PLK1-dependent phosphoproteome analysis by Santamaria et al (2011).
2. The data presented in Fig. 5 are convincing. However, the scheme (Fig. 5j) for measuring chromosome alignment (Fig. 5k) is quite artificial. Why not performing live cell analysis with for example SiR-DNA as conformation of the phenotype?
3. The authors should analyse the interactions of kinetochores with microtubules in T207A and WT cells. Bipolar attachments should be defective in the mutant.
4. The model in Fig. 8b suggests kinetochore-microtubule attachment defects. This defect is consistent with a reduction of BUBR1 and PP2A at kinetochores. How do the authors explain the chromosome bridges in Fig. 6d?
5. Is PP2A reduced on kinetochores in T207A cells?

Minor points

1. Line 189: "Strikingly, we observed no difference in PLK1 binding between ...". Is this also the case when BRCA2 fragments that were expressed in E.coli? Are S206 and T207 phosphorylated in the recombinant BRCA2 fragment or are other P-sites responsible for the phosphatase effect?

Reviewer #1 (Remarks to the Author):

The manuscript is novel, very exciting, well-written, technically sound, a piece of work of high quality and should be considered for publication. The authors identified a direct mitotic role of BRCA2 in the alignment of chromosomes separate from its well-studied role in DNA repair function with direct consequences on chromosome stability. Thus, it is very interesting for a broad readership. For me as a non-expert in mitosis the manuscript was very clear and its biological implications well understandable. The length of the text is appropriate.

Reviewer #2 (Remarks to the Author):

In this manuscript that nicely combines structural and function studies, Ehlén et al. provide strong evidence that phosphorylation of DNA damage response protein BRCA2 by Polo-like kinase PLK1 plays a critical and unexpected role in regulating mitosis. Using multiple approaches, it is shown that two sites in BRCA2, S193 and T207, are phosphorylated and that recognition of pT207 by PLK1 Polo-box domain primes the interaction of PLK1 with BRCA2. Of note, the T207A and S206C variants have been found in breast cancer patients and were shown by the authors to cause defective chromosome congression during mitosis and to delay mitosis progression. The NMR approach used to

unambiguously identify the phosphorylated BRCA2 sites in vitro is particularly exciting. The interaction of PLK1 with pT207 BRCA2 was quantified by calorimetry and X-ray crystallography. Using biochemistry and cell biology approaches, it is shown that the PLK1/BRCA2 complex associates with PP2A and phosphorylated BUBR1 and that the ensuing tetrameric protein assembly contributes to proper chromosome alignment during mitosis. Importantly, data presented in the manuscript suggest that the PLK1/BRCA2 interaction does not play a role in the well-established homologous recombination DNA repair function of BRCA2. While this reviewer is less qualified to comment on the cell biology aspects of the study, the work is clearly of high quality. The conclusions are well supported by the data presented in the manuscript. The manuscript is well organized and well written.

Reviewer #3 (Remarks to the Author):

Proper chromosome alignment depends on BRCA2 phosphorylation by PLK1

Ehlen et al.

The authors address in this manuscript whether phosphorylation of the tumour suppressor BRCA2 by PLK1 is important for mitosis. The authors start by asking whether missense variants in BRCA2 from cancer patients are phosphorylated differently by PLK1 compared to WT BRCA2. Indeed, they find that T207A and S206C in BRCA2 decrease PLK1 phosphorylation in vitro two-fold. Next, the authors use NMR in order to quantify PLK1 phospho-sites in BRCA2 fragments. This analysis shows that the kinetics of S193 phosphorylation is dependent on T207. The authors test the model that T207 is important for the shelf-priming of PLK1 binding. In agreement with this model, T207A and S206C mutant fragments were less efficient in pulling down PLK1 than the phosphorylated WT peptide. Next they analyse the binding of the BRCA2 peptide to the PLK1PBD by crystallography providing a structural understanding of this interaction.

The authors then study the consequences of BRCA2 phosphorylation by PLK1 in vivo using a BRCA2 deficient human cell line. They provide evidence of an additional PLK1 binding site that is primed by CDK1 phosphorylation of T77. Interestingly, the duration of mitosis for S206C and T207A cells is longer than for WT. This mitotic delay was due to a reduction in the formation of a complex between BRCA2, PLK1, BUBR1 and PP2A in the S206C and T207A mutants. In addition, pBUBR1 at kinetochores was reduced in T207A cells. This defect accumulated in chromosome missegregation errors, mostly in lagging chromosomes and in high numbers of aneuploidy. Finally, the authors convincingly show that the delay in mitosis is not an indirect consequence of

DNA damage and that DNA repair was not grossly affected by BRCA2-T207A.

Taken together, this is an interesting manuscript of overall high quality that describes an important function of BRCA2 in complex with PP2A, BUBR1 and PLK1 in mitosis. Disturbing this function of BRCA2, probably in regulating kinetochore-microtubule interactions, leads to chromosome missegregation without having a strong impact on the function of BRCA2 in DNA repair. I support publication of this manuscript in Nat. Comm. after revision.

Specific points

1. What are the evidences that T207 is phosphorylated by PLK1 *in vivo*? Was this P-site identified in Plk1-dependent phosphoproteome analysis by Santamaria et al (2011).

We tried to use phospho-proteomics, however no peptide containing BRCA2 T207 could be detected by MS. This is also the case for T77 phosphorylation, which is well established (Yata et al., 2012). The reason for this is that the protease sites are too far from these phosphosites. To confirm this phosphorylation takes place in cells we have raised a polyclonal antibody against a peptide encompassing pT207. Using this antibody in BRCA2 WT nocodazole-arrested versus asynchronous cells in a Western blotting we observe a band that corresponds to the size of BRCA2 only in the mitotic cell extracts (New Fig. 4a). In addition, phosphatase treatment abolished the signal indicating that the band corresponds to a phosphorylation event. Finally, cells bearing BRCA2-T207A show reduced signal of this antibody while they are positive for BRCA2 antibody. These results indicate that the phosphorylation of T207 takes place *in vivo*. We have now included these data in Fig. 4a.

2. The data presented in Fig. 5 are convincing. However, the scheme (Fig. 5j) for measuring chromosome alignment (Fig. 5k) is quite artificial. Why not performing live cell analysis with for example SiR-DNA as conformation of the phenotype?

We have contemplated this possibility however siR-DNA can induce DNA damage and alter cell cycle progression which might compromise the results (Sen et al., Scientific Reports 2018). To avoid these concerns, we decided not to include these data. We estimate that although complementary, these experiments are not essential for the manuscript as we show a clear mitotic delay (new Fig 4d-f) and misalignment defect (New Fig 6a,b) in the mutated cells. The experiments with Monastrol + MG132 are well documented and has been used to show the misalignment defects observed due to an impaired BUBR1 phosphorylation (see for example Elowe et al., 2007 Genes & Dev). The addition of the double thymidine block was only used to synchronize the cells so that we enrich the population of mitotic cells and therefore count a sufficient number of events to be confident of the results (in our case between 400-900 cells counted per condition).

3. The authors should analyse the interactions of kinetochores with microtubules in T207A and WT cells. Bipolar attachments should be defective in the mutant.

As requested, we have examined the stability of kinetochores-MT interactions in BRCA2 WT vs T207A mutated cells using cold treatment as described before (Lampson and Kapoor

2005; Elowe et al., 2007). Using the same protocol as for the visualization of chromosome alignment, after Monastrol washout and MG132 treatment cells were incubated on ice for 15 min followed by fixation and IF using antibodies against α -tubulin and CREST. As predicted by this reviewer, cold treatment resulted in a net destabilization of microtubules T207A mutated cells compared to the BRCA2 WT cells as measure by the intensity of α -tubulin at the kinetochores indicating that there is indeed a defect of microtubule attachment. These results are now in new Fig. 6c, d.

4. The model in Fig. 8b suggests kinetochore-microtubule attachment defects. This defect is consistent with a reduction of BUBR1 and PP2A at kinetochores. How do the authors explain the chromosome bridges in Fig. 6d?

We observe a considerable number of chromosome bridges already in the cells complemented with BRCA2 (BRCA2 WT) probably due to the cancerous nature of the cells (p53 mutated). BRCA2 deficient cells (BRCA2^{-/-}) show an increased number of chromosome bridges as expected from the DNA repair defect in these cells. However, when comparing T207A and S206C to the BRCA2^{-/-} cells (new Fig. 6a); T207A and S206C rescue most of chromosome bridges observed in BRCA2^{-/-} cells whereas they show an increased number of lagging chromosomes compared to BRCA2^{-/-}. These results support the idea that the HR and mitotic functions of BRCA2 can be uncoupled and that these mutations affect primarily the latter.

5. Is PP2A reduced on kinetochores in T207A cells?

Although we have tried, we have not been able to detect PP2A (or B56 alpha) at the kinetochores with the antibodies at hand therefore, we are unable to answer this question. However, our results show a ~25% reduction of pBUBR1-T680 at the kinetochores in cells bearing T207A variant (Fig. 5 h, i). Because these cells show reduced interaction between PP2A and BUBR1 (fig. 5f) and we know this interaction takes place at the kinetochore (Suijkerbuijk et al., 2012 Dev. Cell) we have indirect evidence that PP2A is either reduced or not fully active at the kinetochore in the mutated cells.

Minor points

1. Line 189: "Strikingly, we observed no difference in PLK1 binding between ...". Is this also the case when BRCA2 fragments that were expressed in E.coli?

We understand that there is a possibility that the fragment purified from human cells is already phosphorylated. To resolve this issue we have not performed this experiment with BRCA2 fragments produced in bacteria but we did the following experiment that answers the same question. We pre-incubated BRCA2 1-250-WT with phosphatase before the addition of PLK1 and performed a pull-down assay. In these conditions, there was a 2-fold decrease in the binding to PLK1 indicating that the phosphorylation of BRCA2 is required/favors the interaction with PLK1 (Fig. 3f lane 12 compared to 10, Fig. 3g). We also know that a 17 aa synthesized peptide comprising pT207 is sufficient to bind PLK1-Polo Box Domain with an affinity of 90 nM whereas the same peptide non-phosphorylated at T207 cannot bind (Fig. 3h, i). We have also mapped the specific interaction site with the polo-box domain of PLK1 in the 3D structure (Fig. 3l).

Are S206 and T207 phosphorylated in the recombinant BRCA2 fragment or are other P-sites responsible for the phosphatase effect?

S206 is not phosphorylated by PLK1. By NMR we identified T207 as phosphorylated by PLK1 in the recombinant fragment together with other residues such as S193 (Fig. 2). We cannot discard that other phosphorylation sites also favor the interaction with T207 but the fact that T207A mutation reduces very much the binding (Fig. 3c, d) suggests that pT207 favors the interaction.

REVIEWERS' COMMENTS:

Reviewer #3 (Remarks to the Author):

The authors have addressed all points that I have raised. I am very happy with the revision of the manuscript. It is now ready for publication in Nat Comm.

Point by point response to reviewers comments

REVIEWERS' COMMENTS:

Reviewer #3 (Remarks to the Author):

The authors have addressed all points that I have raised. I am very happy with the revision of the manuscript. It is now ready for publication in Nat Comm.

Thank you for your help.